# 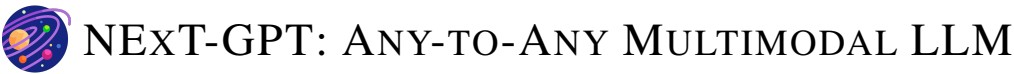 NExT-GPT: Any-to-Any Multimodal LLM

## Abstract

While recently Multimodal Large Language Models (MM-LLMs) have made exciting strides, they mostly fall prey to the limitation of only input-side multimodal understanding, without the ability to produce content in multiple modalities. As we humans always perceive the world and communicate with people through various modalities, developing any-to-any MM-LLMs capable of accepting and delivering content in any modality becomes essential to human-level AI. To fill the gap, we present an end-to-end general-purpose any-to-any MM-LLM system, **NExT-GPT**. We connect an LLM with multimodal adaptors and different diffusion decoders, enabling NExT-GPT to perceive inputs and generate outputs in arbitrary combinations of text, image, video, and audio. By leveraging the existing well-trained high-performing encoders and decoders, NExT-GPT is tuned with only a small amount of parameter (1%) of certain projection layers, which not only benefits low-cost training but also facilitates convenient expansion to more potential modalities. Moreover, we introduce a modality-switching instruction tuning (MosIT) and manually curate a high-quality dataset for MosIT, based on which NExT-GPT is empowered with complex cross-modal semantic understanding and content generation. Overall, our research showcases the promising possibility of building a unified AI agent capable of modeling universal modalities, paving the way for more human-like AI research in the community.

## 1 Introduction

Recently, the topic of Artificial Intelligence Generated Content (AIGC) has witnessed unprecedented advancements with certain technologies, such as ChatGPT for text generation (OpenAI, 2022a) and diffusion models for visual generation (Fan et al., 2022). Among these, the rise of Large Language Models (LLMs) has been particularly remarkable, e.g., Flan-T5 (Chung et al., 2022), Vicuna (Chiang et al., 2023), LLaMA (Touvron et al., 2023) and Alpaca (Taori et al., 2023), showcasing their formidable human-level language reasoning and decision-making capabilities, shining a light on the path of Artificial General Intelligence (AGI). Our world is inherently multimodal, and humans perceive the world with different sensory organs for varied modal information, such as language, images, videos, and sounds, which often complement and synergize with each other. With such intuition, the purely text-based LLMs have recently been endowed with other modal understanding and perception capabilities of image, video, audio, etc.

A notable approach involves employing adapters that align pre-trained encoders in other modalities to textual LLMs. This endeavor has led to the rapid development of multimodal LLMs (MM-LLMs), such as BLIP-2 (Li et al., 2023c), Flamingo (Alayrac et al., 2022), MiniGPT-4 (Zhu et al., 2023), Video-LLaMA (Zhang et al., 2023c), LLaVA (Liu et al., 2023b), PandaGPT (Su et al., 2023), and SpeechGPT (Zhang et al., 2023b). Nevertheless, most of these efforts pay attention to the multimodal content understanding at the input side, while lacking the ability to output content in multiple modalities other than texts. We emphasize that natural human cognition and communication indispensably require seamless transitions between any modalities of information. This makes the exploration of any-to-any MM-LLMs critical to achieving real AGI, i.e., the ability to accept inputs in any modality and deliver responses in any appropriate modality.

Certain efforts have been made to mimic the human-like any-to-any modality conversion. Lately, CoDi (Tang et al., 2023) has made strides in implementing the capability of simultaneously processing and generating arbitrary combinations of modalities; however, it lacks the reasoning and decision-making prowess of LLMs as its core, and is also limited to the simple paired content generation. On the other

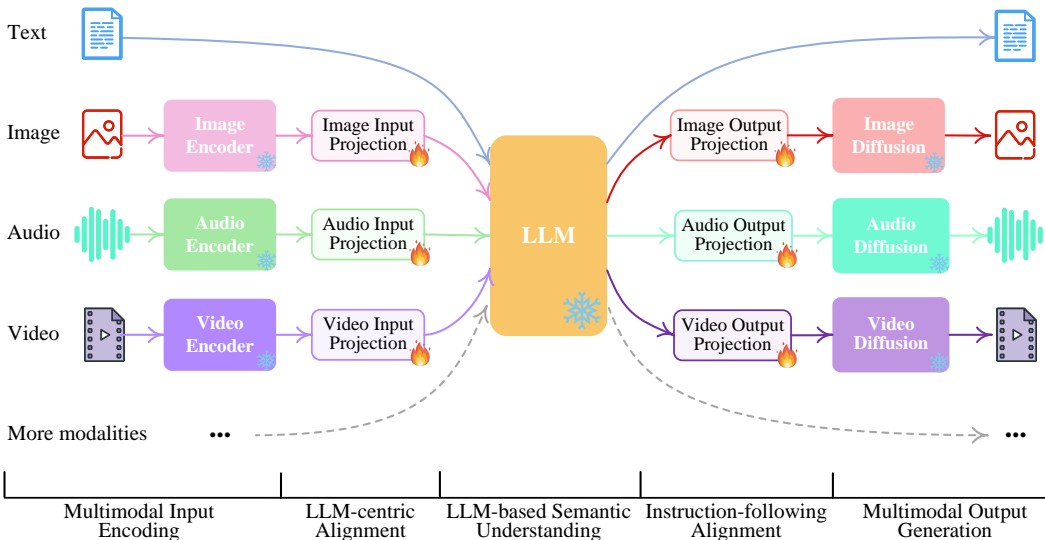

Figure 1: By connecting LLM with multimodal adaptors and diffusion decoders, NExT-GPT achieves universal multimodal understanding and any-to-any modality input and output, with ❄representing the frozen module and 🔥denoting the trainable module.

hand, some efforts, e.g., Visual-ChatGPT (Wu et al., 2023) and HuggingGPT (Shen et al., 2023), have sought to combine LLMs with external tools to achieve approximately the 'any-to-any' multimodal understanding and generation. Unfortunately, these systems suffer from critical challenges due to their complete pipeline architecture. First, the information transfer between different modules is entirely based on discrete texts produced by the LLM, where the cascading process inevitably introduces noise and propagates errors. More critically, the entire system leverages existing pre-trained tools for inference only. Due to the lack of overall end-to-end training, the capabilities of content understanding and multimodal generation can be very limited, especially in interpreting intricate and implicit user instructions. In a nutshell, there is a compelling need for constructing an end-to-end MM-LLM of arbitrary modalities.

In pursuit of this goal, we present **NExT-GPT**, an any-to-any MM-LLM designed to seamlessly handle input and output in any combination of four modalities: text, image, video, and audio. As depicted in Figure 1, NExT-GPT comprises three tiers. **First**, we leverage established encoders to encode inputs in various modalities, where these representations are projected into language-like representations comprehensible to LLM through a projection layer. **Second**, we harness an existing open-sourced LLM as the core to process input information for semantic understanding and reasoning. The LLM not only directly generates text tokens but also produces unique "modality signal" tokens that serve as instructions to dictate the decoding layers on whether and what modal content to output correspondingly. **Third**, after projection, the produced multimodal signals with specific instructions are routed to different encoders and finally generate content in corresponding modalities.

As NExT-GPT encompasses encoding and generation of various modalities, training the system from scratch would entail substantial costs. Instead, we take advantage of the existing pre-trained high-performance encoders and decoders, such as ViT (Dosovitskiy et al., 2021), ImageBind (Girdhar et al., 2023) and the state-of-the-art latent diffusion models (Rombach et al., 2022; Ruiz et al., 2022; Cerspense, 2023; An et al., 2023; Liu et al., 2023a; Huang et al., 2023a). By loading the off-the-shelf parameters, we not only avoid cold-start training but also facilitate the potential growth of more modalities. For feature alignment across the three tiers, we only consider fine-tuning locally the input projection and output projection layers, with an encoding-side LLM-centric alignment and decoding-side instruction-following alignment, where the minimal computational overhead ensures higher efficiency. Furthermore, to empower our any-to-any MM-LLM with human-level capabilities in complex cross-modal generation and reasoning, we introduce a *modality-switching instruction tuning*, to equip the system with sophisticated cross-modal semantic understanding and content generation. To combat the absence of such cross-modal instruction tuning data in the community, we manually collect and annotate a MosIT dataset consisting of 5,000 high-quality samples. By

employing the LoRA technique (Hu et al., 2022), we fine-tune the overall NExT-GPT system on instruction tuning data, updating both input and output projection layers and certain LLM parameters.

Overall, this work showcases the promising possibility of developing a more human-like MM-LLM agent capable of modeling universal modalities. The contributions of this paper are as follows:

- We, for the first time, present an end-to-end general-purpose any-to-any MM-LLM, named NExT-GPT, capable of semantic understanding and reasoning and generation of free input and output combinations of text, image, video, and audio.
- We introduce lightweight alignment learning techniques, the LLM-centric alignment at the encoding side, and the instruction-following alignment at the decoding side, efficiently requiring only minimal parameter adjustments (only 1% params) for effective semantic alignment.
- We annotate a high-quality modality-switching instruction tuning dataset covering intricate instructions across various modal combinations of text, image, video, and audio, aiding MM-LLM with human-like cross-modal content understanding and instruction reasoning.

## 2 RELATED WORK

**Cross-modal Understanding and Generation** Our world is replete with multimodal information, wherein we continuously engage in the intricate task of comprehending and producing cross-modal content. The AI community correspondingly emerges varied forms of cross-modal learning tasks (Zeng et al., 2023; Dessì et al., 2023; Yang et al., 2021; Ding et al., 2021; Liu et al., 2023a; Dorkenwald et al., 2021). Moreover, to generate high-quality content, a multitude of strong-performing methods have been proposed, such as Transformer (Vaswani et al., 2017; Zhang et al., 2022; Ding et al., 2021; Ge et al., 2022), GANs (Liu et al., 2020; Brock et al., 2019; Xu et al., 2018; Zhu et al., 2019), VAEs (Vahdat & Kautz, 2020; Razavi et al., 2019), Flow models (Shibata et al., 2022; Bashiri et al., 2021) and the current state-of-the-art diffusion models (Hoogeboom et al., 2021; Qu et al., 2023b; Mou et al., 2023; Feng et al., 2022; Rombach et al., 2022). In particular, the diffusion-based methods have recently delivered a remarkable performance in a plethora of cross-modal generation tasks, such as DALL-E (Ramesh et al., 2021), Stable Diffusion (Rombach et al., 2022). While all previous efforts of cross-modal learning are limited to the comprehension of multimodal inputs only, CoDi (Tang et al., 2023) lately presents groundbreaking development. Leveraging the power of diffusion models, CoDi possesses the ability to generate any combination of output modalities, including language, image, video, or audio, from any combination of input modalities in parallel. Regrettably, CoDi still falls short of achieving human-like deep reasoning of input content, because it can only deliver parallel cross-modal feeding&generation without any reasoning and decision-marking capabilities.

**Multimodal Large Language Models** LLMs have already made a profound impact and revolution on the entire AI community and beyond (OpenAI, 2022a;b), where a series of open-source LLMs have greatly spurred advancement and made contributions to the community (Chiang et al., 2023; Touvron et al., 2023; Zhu et al., 2023; Zhang et al., 2023a). Building on top of these LLMs, significant efforts have been made to extend them to deal with multimodal inputs and tasks, leading to the development of MM-LLMs. On the one hand, most of researchers build fundamental MM-LLMs by aligning the well-trained encoders of various modalities to the textual feature space of LLMs to perceive other modal inputs (Huang et al., 2023c; Zhu et al., 2023; Su et al., 2022; Koh et al., 2023). For example, Flamingo (Alayrac et al., 2022) uses a cross-attention layer to connect a frozen image encoder to the LLMs. BLIP-2 (Li et al., 2023c) employs a Q-Former to translate the input image queries to the LLMs. There are also various similar practices for building MM-LLMs that are able to understand video (e.g., Video-Chat (Li et al., 2023d) and Video-LLaMA (Zhang et al., 2023c)), audio (e.g., SpeechGPT (Zhang et al., 2023b)), etc. Profoundly, PandaGPT (Su et al., 2023) achieves a comprehensive understanding of six different modalities simultaneously by integrating the multimodal encoder, i.e., ImageBind (Girdhar et al., 2023).

Nevertheless, these MM-LLMs are all limited to the limitation of only perceiving multimodal data, without the ability to generate content in arbitrary modalities. To enable LLMs with both multimodal input and output, some efforts explore employing LLMs as decision-makers, and utilizing existing off-the-shelf multimodal encoders and decoders as tools to execute multimodal input and output, such as Visual-ChatGPT (Wu et al., 2023), HuggingGPT (Shen et al., 2023), and AudioGPT (Huang et al., 2023b). As aforementioned, passing messages between modules with pure texts (i.e., LLM textual instruction) under the discrete pipeline scheme will inevitably introduce noises. Also, the

| | Encoder | | Input Projection | | LLM | | Output Projection | | Diffusion | |
|---|---|---|---|---|---|---|---|---|---|---|
| | Name | Param | Name | Param | Name | Param | Name | Param | Name | Param |
| **Text** | — | — | — | — | | | — | — | — | — |
| **Image** | | | | | Vicuna | 7B ❄ | Transformer | 31M🔥 | SD | 1.3B ❄ |
| **Audio** | ImageBind | 1.2B ❄ | Linear | 4M🔥 | (LoRA | 33M🔥) | Transformer | 31M🔥 | AudioLDM | 975M ❄ |
| **Video** | | | | | | | Transformer | 32M🔥 | Zeroscope | 1.8B ❄ |

Table 1: Summary of system configuration. Only 1% of parameters need updating.

lack of comprehensive tuning across the whole system significantly limits the efficacy of semantics understanding. Our work takes the mutual benefits of both the above two types, i.e., learning an any-to-any MM-LLM in an end-to-end manner.

## 3 OVERALL ARCHITECTURE

Figure 1 presents the schematic overview of the NExT-GPT framework. It consists of three main tiers: the encoding stage, the LLM understanding and reasoning stage, and the decoding stage.

**Multimodal Encoding Stage** First, we leverage existing well-established models to encode inputs of various modalities. There are a set of alternatives of encoders for different modalities, e.g., Q-Former (Li et al., 2023c), ViT (Dosovitskiy et al., 2021), CLIP (Radford et al., 2021), HuBERT (Hsu et al., 2021). Here we take advantage of the ImageBind (Girdhar et al., 2023), which is a unified high-performance encoder across six modalities. With ImageBind, we are spared from managing many numbers of heterogeneous modal encoders. Then, via the linear projection layer, different input representations are mapped into language-like representations that are comprehensible to the LLM.

**LLM Understanding and Reasoning Stage** An LLM is used as the core agent of NExT-GPT. Technically, we employ the Vicuna[1] (Chiang et al., 2023), which is the open-source text-based LLM that is widely used in the existing MM-LLMs (Su et al., 2023; Zhang et al., 2023c). LLM takes as input the representations from different modalities and carries out semantic understanding and reasoning over the inputs. It outputs: 1) the textual responses directly, and 2) signal tokens of each modality that serve as instructions to dictate the decoding layers on whether to generate multimodal contents and what content to produce if yes.

**Multimodal Generation Stage** Receiving the multimodal signals with specific instructions from LLM (if any), the Transformer-based output projection layers map the signal token representations into the ones that are understandable to the following multimodal decoders. Technically, we employ the current off-the-shelf latent conditioned diffusion models of different modal generations, i.e., Stable Diffusion (SD)[2] for image synthesis (Rombach et al., 2022), Zeroscope[3] for video synthesis (Cerspense, 2023), and AudioLDM[4] for audio synthesis (Liu et al., 2023a). After a projection layer, the signal representations are fed into the conditioned diffusion models for content generation.

In Table 1 we summarize the overall system configurations. It is noteworthy that in the entire system, only the input and output projection layers of lower-scale parameters (compared with the overall huge capacity framework) are required to be updated during the following learning, with all the rest of the encoders and decoders frozen. This amounts to, 131M(=4+33+31+31+32) / [131M + 12.275B(=1.2+7+1.3+1.8+0.975)], or only **1%** of parameters need to be updated. This is also one of the key advantages of our MM-LLM.

## 4 LIGHTWEIGHT MULTIMODAL ALIGNMENT LEARNING

To bridge the gap between the feature space of different modalities, and ensure fluent semantics understanding of different inputs, it is essential to perform alignment learning for NExT-GPT. Since we design the loosely-coupled system with mainly three tiers, we only need to update the two projection layers at the encoding side and decoding side.

---

[1] https://huggingface.co/lmsys/vicuna-7b-delta-v0, 7B, version 0.

[2] https://huggingface.co/runwayml/stable-diffusion-v1-5, version 1.5.

[3] https://huggingface.co/cerspense/zeroscope_v2_576w, version zeroscope_v2_576w.

[4] https://audioldm.github.io/, version audioldm-l-full.

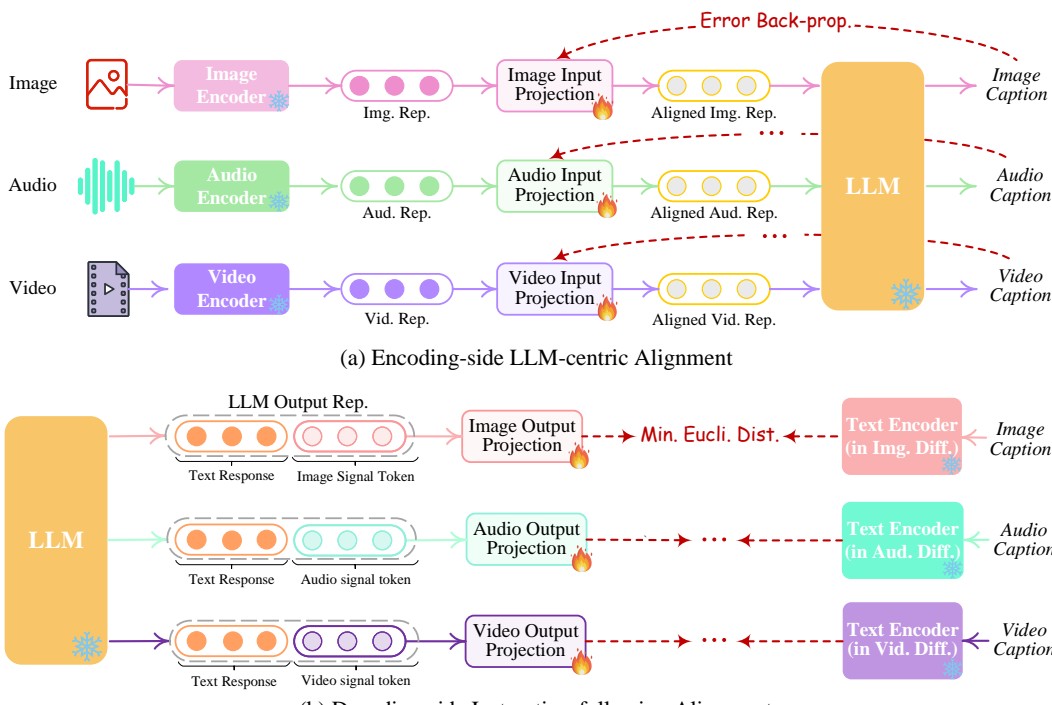

(a) Encoding-side LLM-centric Alignment

(b) Decoding-side Instruction-following Alignment

Figure 2: Illustration of the lightweight multimodal alignment learning of encoding and decoding.

## 4.1 ENCODING-SIDE LLM-CENTRIC MULTIMODAL ALIGNMENT

Following the common practice of existing MM-LLMs, we consider aligning different inputting multimodal features with the text feature space, resulting in representations that are understandable to the core LLM. This is intuitively named the LLM-centric multimodal alignment learning. To accomplish the alignment, we adopt an 'X-to-text' generation task trained on the 'X-caption' pair ('X' stands for image, audio, or video) data from existing corpus and benchmarks, i.e., given the representation of an 'X', to prompt the frozen LLM to generate the corresponding text description. Specifically, we utilize three types of 'X-caption' pair data, including: 1) 'Video-caption' pair dataset: Webvid-2M (Bain et al., 2021), a large-scale dataset of short videos with textual description sourced from stock footage sites, 2) 'Image-caption' pair dataset: CC3M (Sharma et al., 2018), contains over 3 million images accompanied by diverse styles of natural-language descriptions, and 3) 'Audio-caption' pair dataset: AudioCaps (Kim et al., 2019), an extensive dataset of approximately 46k audio clips paired with human-written textual descriptions collected via crowdsourcing. Figure 2(a) illustrates the learning process.

## 4.2 DECODING-SIDE INSTRUCTION-FOLLOWING ALIGNMENT

On the decoding end, we have integrated pre-trained conditional diffusion models from external resources. Our main purpose is to align the diffusion models with LLM's output instructions. However, performing a full-scale alignment process between each diffusion model and the LLM would entail a significant computational burden. Alternatively, we explore a more efficient approach, decoding-side instruction-following alignment, as depicted in Figure 2(b). Specifically, instead of outputting straightforward textual instructions, we design three types of special tokens (Koh et al., 2023), i.e., '$[IMG_i]$' ($i = 0, \cdots, 4$) as image signal tokens; '$[AUD_i]$' ($i = 0, \cdots, 8$) as audio signal tokens; and '$[VID_i]$' ($i = 0, \cdots, 24$) as video signal tokens; these tokens implicitly carry rich and flexible instructions for the downstream diffusion model. We want to enable the LLM to learn what content to generate, i.e., textual tokens, and modality signal tokens. If LLM identifies a certain modality content (except language) to be produced, a special type of token will be output indicating the activation of that modality; otherwise, no special token output means deactivation of that modality. We notice that diffusion models of various modalities are conditioned solely on the representations extracted from the text encoders in different modal diffusion models. However, this conditioning diverges significantly from the modal signal tokens from LLM in our system. This leads to a gap

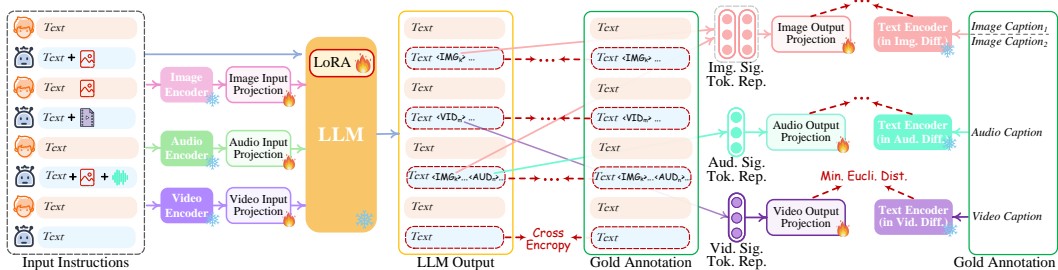

Figure 3: Illustration of modality-switching instruction tuning.

that prevents the diffusion models from accurately interpreting the instructions from LLM. Thus, we consider minimizing the distance between the LLM's modal signal token representations (after each Transformer-based project layer) and the conditional text representations of the diffusion models. Since only the textual condition encoders are used (with the diffusion backbone frozen), the learning is merely based on the purely captioning texts, i.e., without any visual or audio resources. This also ensures a highly lightweight training. Technically, to endow the model to produce other modalities beyond text, we add the signal tokens to the vocabulary of the LLM. In the alignment training phase, we take the captions from CC3M, WebVid, and AudioCaps as inputs and concatenate them with the signal tokens as outputs. The loss function comprises two key components: 1) the negative log-likelihood of producing signal tokens, and 2) the $l_2$-distance between the hidden states of signal tokens produced by the LLM and the conditional text representations derived from the text encoder within diffusion models.

## 5 MODALITY-SWITCHING INSTRUCTION TUNING

### 5.1 INSTRUCTION TUNING

Despite aligning both the encoding and decoding ends with LLM, there remains a gap towards the goal of enabling the overall system to faithfully follow and understand users' instructions and generate the desired multimodal outputs. To address this, further instruction tuning (IT) (Yin et al., 2023; Su et al., 2023; Liu et al., 2023b) is deemed necessary to enhance the capabilities and controllability of LLM. IT involves additional training of overall MM-LLMs using '*(INPUT, OUTPUT)*' pairs, where '*INPUT*' represents the user's instruction, and '*OUTPUT*' signifies the desired model output that conforms to the given instruction. Technically, we leverage LoRA (Hu et al., 2022) to enable a small subset of parameters within NExT-GPT to be updated concurrently with two layers of projection during the IT phase. As illustrated in Figure 3, when an IT dialogue sample is fed into the system, the LLM reconstructs and generates the textual content of input (and represents the multimodal content with the multimodal signal tokens). The optimization is imposed based on gold annotations and LLM's outputs. In addition to LLM tuning, we also fine-tune the decoding end of NExT-GPT. We align the modal signal tokens' representation encoded by the output projection with the gold multimodal caption representation encoded by the diffusion condition encoder. Thereby, the comprehensive tuning process brings closer to the goal of faithful and effective interaction with users.

### 5.2 INSTRUCTION DATASET

For the IT of NExT-GPT, we consider the following datasets:

**'Text+X' → 'Text' Datasets**   The commonly used datasets for MM-LLM IT entail inputs of both texts and multimodal contents (i.e., 'X' could be the image, video, audio, or others), and the outputs are textual responses from LLM. There are well-established datasets, e.g., LLaVA (Liu et al., 2023b), miniGPT-4 (Zhu et al., 2023), VideoChat (Li et al., 2023d), where we directly employ them for our tuning purpose.

**'Text' → 'Text+X' Datasets**   Significantly unlike existing MM-LLMs, in our any-to-any scenario, the target not only includes the generations of texts, but also the multimodal contents, i.e., 'Text+X'. Thus, we construct the 'Text' → 'Text+X' dataset, i.e., text-to-multimodal (namely T2M) data. Based on the rich volume of 'X-caption' pairs from the existing corpus and benchmarks (Sharma et al., 2018; Lin et al., 2014; Bain et al., 2021; Kim et al., 2019), with some templates, we employ GPT-4 to produce varied textual instructions to wrap the captions, and result in the dataset.

**MosIT Dataset** Crafting high-quality instructions that comprehensively cover the desired target behaviors is non-trivial. We notice that the above IT datasets fail to meet the requirements for our any-to-any MM-LLM scenario. Firstly, during a human-machine interaction, users and LLM involve diverse and dynamically changing modalities in their inputs and outputs. Additionally, we allow multi-turn conversations in the process, and thus the processing and understanding of complex user intentions is required. However, the above two types of datasets lack variable modalities, and also are relatively short in dialogues, failing to mimic real-world scenarios adequately.

To facilitate the development of any-to-any MM-LLM, we propose a novel Modality-switching Instruction Tuning (MosIT) approach. MosIT not only supports complex cross-modal understanding and reasoning but also enables sophisticated multimodal content generation. In conjunction with MosIT, we manually and meticulously construct a high-quality dataset. The MosIT dataset encompasses a wide range of multimodal inputs and outputs, offering the necessary complexity and variability to facilitate the training of MM-LLMs that can handle diverse user interactions and deliver the desired responses accurately. Specifically, we design some template dialogue examples between a 'Human' role and a 'Machine' role, based on which we prompt the GPT-4 to generate more conversations under various scenarios with more than 100 topics or keywords. The interactions are required to be diversified, e.g., including both straightforward and implicit requirements by the 'Human', and execution of perception, reasoning, suggestion, and planning, etc., by the 'Machine'. And the interactive content should be logically connected and semantically inherent and complex, with in-depth reasoning details in each response by the 'Machine'. Each conversation should include 3-7 turns (i.e., QA pairs), where the 'Human'-'Machine' interactions should involve multiple modalities at either the input or output side, and switch the modalities alternately. Whenever multimodal contents (e.g., image, audio, and video) are present in the conversations, we look for the best-matched contents from the external resources, including the retrieval systems, e.g., Youtube[5], and even AIGC tools, e.g., Stable-XL (Podell et al., 2023), and Midjourney[6]. After human inspections and filtering of inappropriate instances, we obtain a total of 5K high-quality dialogues. In Table 15 of Appendix §I, we compare the statistics of existing multimodal IT datasets with our MosIT data in detailed statistics.

## 6 EXPERIMENTS

Here we quantify the capability of NExT-GPT across different cross-modal learning tasks, including text-to-'X' generation, 'X'-to-text generation, and Text-conditioned modality editing. We mimic the task by taking only one turn of interaction between the user and the model. To align with existing works, we consider five frequently-adopted benchmarks, including three 'Text-X' pair datasets: 1) COCO-caption (Lin et al., 2014), 2) MSR-VTT (Xu et al., 2016), and 3) AudioCaps (Kim et al., 2019); as well as two text-conditioned 'X' editing dataset: 4) VCTK (Veaux et al., 2017) and 5) DAVIS (Perazzi et al., 2016). We compare our system with the best-performing baseline models across various tasks. To ensure a fair comparison, we adhere to the experimental settings used in the baselines of each dataset, including the data splitting and fine-tuning/zero-shot setups. We employ the following metrics to assess the quality of generated images, audio, and video: FID (Heusel et al., 2017), IS (Salimans et al., 2016), CLIP (Hessel et al., 2021). Furthermore, for text generation, we utilize BLEU (Papineni et al., 2002), METEOR (Denkowski & Lavie, 2014), SPIDEr (Liu et al., 2017), and CIDEr (Vedantam et al., 2015) scores. Due to space limitation, more details of datasets utilized for the training and evaluation of NExT-GPT can be found in Appendix §D, and model training steps in Appendix §C.

**'Text' → 'X' Generation** We first examine the synthesis quality of the image, video, or audio conditioned on text. Table 2, 4, 3, and 11 present the comparisons between ours and some state-of-the-art systems. On text-to-image and text-to-audio generation tasks, NExT-GPT shows a nice performance on par that of the best-performing baselines. Notably, under the zero-shot setting, NExT-GPT shows a significant superiority in video generation conditioning on text, demonstrating the remarkable generalization capability of NExT-GPT.

**'X' → 'Text' Generation** We evaluate the NExT-GPT on the tasks of textual caption generation to test the semantic understanding capability w.r.t. image, video, or audio. The results on different tasks are shown in Table 6, 5, and 7. Significantly, NExT-GPT mostly achieves much better performance

---

[5]https://www.youtube.com/
[6]https://www.midjourney.com/

| Method | FID (↓) |
|---|---|
| CogView (Ding et al., 2021) | 27.10 |
| GLIDE (Nichol et al., 2022) | 12.24 |
| CoDi (Tang et al., 2023) | 11.26 |
| SD (Rombach et al., 2022) | **11.21** |
| NExT-GPT | 11.28 |

Table 2: Text-to-image generation results on COCO-caption (Lin et al., 2014).

| Method | FID (↓) | CLIPSIM (↑) |
|---|---|---|
| CogVideo (Hong et al., 2022) | 23.59 | 0.2631 |
| MakeVideo (Singer et al., 2022) | 13.17 | 0.3049 |
| Latent-VDM (Rombach et al., 2022) | 14.25 | 0.2756 |
| Latent-Shift (An et al., 2023) | 15.23 | 0.2773 |
| CoDi (Tang et al., 2023) | — | 0.2890 |
| NExT-GPT | **13.04** | **0.3085** |

Table 3: Text-to-video generation results (zero-shot) on MSR-VTT (Xu et al., 2016).

| Method | FD (↓) | IS (↑) |
|---|---|---|
| DiffSound (Yang et al., 2023) | 47.68 | 4.01 |
| AudioLDM-S (Liu et al., 2023a) | 29.48 | 6.90 |
| AudioLDM-L (Liu et al., 2023a) | 23.31 | 8.13 |
| CoDi (Tang et al., 2023) | **22.90** | **8.77** |
| NExT-GPT | 23.58 | 8.35 |

Table 4: Text-to-audio generation results on AudioCaps (Kim et al., 2019).

| Method | SPIDEr | CIDEr |
|---|---|---|
| AudioCaps (Kim et al., 2019) | 0.369 | 0.593 |
| BART (Gontier et al., 2021) | 0.465 | 0.753 |
| AL-MixGen (Kim et al., 2022) | 0.466 | 0.755 |
| CoDi (Tang et al., 2023) | 0.480 | 0.789 |
| NExT-GPT | **0.521** | **0.802** |

Table 5: Audio-to-text generation (audio captioning) results on AudioCaps (Kim et al., 2019).

| Method | B@4 | METEOR | CIDEr |
|---|---|---|---|
| Oscar (Li et al., 2020) | 36.58 | 30.4 | 124.12 |
| BLIP-2 (Li et al., 2023c) | 43.7 | — | 145.8 |
| OFA (Wang et al., 2022b) | **44.9** | 32.5 | 154.9 |
| CoDi (Tang et al., 2023) | 40.2 | 31.0 | 149.9 |
| NExT-GPT | 44.3 | **32.9** | **156.7** |

Table 6: Image-to-text generation (image captioning) results on COCO-caption (Lin et al., 2014).

| Method | B@4 | METEOR |
|---|---|---|
| ORG-TRL (Zhang et al., 2020) | 43.6 | 28.8 |
| GIT (Wang et al., 2022a) | 54.8 | 33.1 |
| mPLUG-2 (Xu et al., 2023) | 57.8 | 34.9 |
| CoDi (Tang et al., 2023) | 52.1 | 32.5 |
| NExT-GPT | **58.4** | **38.5** |

Table 7: Video-to-text generation (video captioning) results on MSR-VTT (Xu et al., 2016).

| Method | Object | | Background | |
|---|---|---|---|---|
| | CLIP (↑) | FID (↓) | CLIP (↑) | FID (↓) |
| PTP (Hertz et al., 2023) | 30.33 | 9.58 | 31.55 | 13.92 |
| BLDM (Avrahami et al., 2023) | 29.95 | 6.14 | 30.38 | 20.44 |
| DiffEdit (Couairon et al., 2023) | 29.30 | **3.78** | 26.92 | **1.74** |
| PFB-Diff (Huang et al., 2023d) | **30.81** | 5.93 | **32.25** | 13.77 |
| NExT-GPT | 29.31 | 6.52 | 27.29 | 15.20 |

Table 8: Text+image-to-image generation (text-conditioned image editing) results on COCO-caption (Lin et al., 2014).

| Method | CLIP-T | CLIP-I |
|---|---|---|
| CogVideo (Hong et al., 2022) | 0.2391 | 0.9064 |
| TuneVideo (Wu et al., 2022) | 0.2758 | 0.9240 |
| SDEdit (Meng et al., 2022) | 0.2775 | 0.8731 |
| Pix2Video (Ceylan et al., 2023) | **0.2891** | **0.9767** |
| NExT-GPT | 0.2683 | 0.9645 |

Table 9: Text+video-to-video generation (text-conditioned video editing) results on DAVIS (Perazzi et al., 2016).

on the X-to-text generation than that of the CoDi baseline, owing to the direct generation of texts from LLM, which is inherently expertized by the LLM. Moreover, as demonstrated in Table 11, our system consistently outperforms other MM-LLMs under a zero-shot setting.

**'Text+X' → 'X' Generation**  We also test our model on a task category of text-conditioned modal editing. Table 8, 10 and 9 show the performances on different tasks. Compared with the above two types of tasks, although NExT-GPT did not demonstrate superior performance on the text-conditioned modal editing tasks, it still shows competitive performance.

**Evaluation on Multimodal LLM Benchmark**  Here, we conduct the experiments on recent multimodal LLM benchmarks, including MME (Fu et al., 2023), MMBench (Liu et al., 2023c), and SEEDBench (Li et al., 2023b), as

| Method | MCD (↓) |
|---|---|
| CampNet (Wang et al., 2022c) | 0.380 |
| MakeAudio (Huang et al., 2023a) | 0.375 |
| AudioLDM-L (Liu et al., 2023a) | 0.349 |
| NExT-GPT | **0.302** |

Table 10: Text+audio-to-audio generation (text-conditioned speech editing) results on VCTK (Veaux et al., 2017).

| Model | Version | Image-to-text Generation | | | Text-to-Image generation |
|---|---|---|---|---|---|
| | | NoCaps | Flickr 30K | COCO | COCO |
| ● **MM-LLMs for Multimodal Comprehension Only** | | | | | |
| InstructBLIP (Dai et al., 2023) | instruct_vicuna7B | 123.1* | 82.4* | 102.2$^{†}$ | - |
| LLaVA (Liu et al., 2023b) | LLaMA-2-7B-Chat | 120.7 | 82.7 | - | - |
| mPLUG-Owl (Ye et al., 2023b) | mPLUG-Owl-7B | 117.0 | 80.3 | 119.3 | - |
| ● **MM-LLMs for Multimodal Comprehension & Generation** | | | | | |
| EMU (Sun et al., 2023) | LLaMA-13B | - | - | 117.7$^{‡}$ | 11.66$^{‡,♮}$ |
| DreamLLM (Dong et al., 2023) | 7B | - | - | 115.4$^{‡}$ | 8.46$^{‡,♮}$ |
| NExT-GPT | Vicuna-7B | **123.6** | **84.5** | **124.9** | 13.85 (8.62$^{♮}$) |

Table 11: Zero-shot evaluation of image-to-text generation with CIDEr (↑) score on NoCaps (Agrawal et al., 2019), Flickr 30K (Young et al., 2014) and COCO (Karpathy & Fei-Fei, 2017) and text-to-image generation with FID (↓) score on COCO. Results marked with * are sourced from Dai et al. (2023), $^{†}$ from Ye et al. (2023c), and $^{‡}$ from Dong et al. (2023). Results marked with $^{♮}$ are from models with additional pre-training on LION data (Schuhmann et al., 2022).

| Model | Coarse-grained Tasks | | | Fine-grained Tasks | | | Reasoning Tasks | | |
|---|---|---|---|---|---|---|---|---|---|
| | Existence | Count | Color | Poster | Celebrity | Scene | Commonsense Reasoning | Numerical Calculation | Text Translation |
| LLaVA(7B)* | 50 | 50.00 | 55.00 | 50.00 | 48.82 | 50.00 | 57.14 | 50.00 | 57.50 |
| InstructBLIP(flant5xxl)* | **185** | **143.33** | 153.33 | 123.81 | 101.18 | 153.00 | **129.29** | 40.00 | 65.00 |
| mPLUG-Owl(7B)* | 120 | 50.00 | 55.00 | **136.50** | 100.29 | 135.50 | 78.57 | 60.00 | **80.00** |
| NExT-GPT(7B) | 180 | 96.67 | **156.67** | 110.00 | **103.00** | **156.25** | 116.14 | **62.50** | 65.50 |

Table 12: Evaluation results (%) on MME for Coarse-Grained, Fine-Grained, and Reasoning Tasks. Results marked with * are sourced from Fu et al. (2023).

| Model | MMBench | | | | | | | SEEDBench | | |
|---|---|---|---|---|---|---|---|---|---|---|
| | Overall | LR | AR | RR | FP-S | FP-C | CP | Overall | Img | Video |
| LLaVA(7B)* | 36.2 | 15.9 | 53.6 | 28.6 | 41.8 | 20.0 | 40.4 | - | - | - |
| InstructBLIP(7B)* | 33.9 | 21.6 | 47.4 | 22.5 | 33.0 | 24.4 | 41.1 | 53.4 | 58.8 | 38.1 |
| mPLUG-Owl(7B)* | 46.6 | 19.9 | 56.1 | **39.0** | **53.0** | 26.8 | 59.4 | 34.0 | 37.9 | 23.0 |
| NExT-GPT(7B) | **48.0** | **22.1** | **60.5** | 33.6 | 46.8 | **30.7** | **60.6** | **54.4** | **59.2** | **39.4** |

Table 13: Evaluation results (%) on MMBench test set (L-2 abilities), and SEEDBench. Results marked with * are sourced from Liu et al. (2023c) and (Li et al., 2023b).

shown in Table 12 and 13. Observing the results, our model mostly achieves better performance than the comparing baseline MM-LLMs.

**Qualitative Results**   To directly demonstrate the effectiveness and potential of NExT-GPT in developing human-like conversational agents, we further offer compelling examples that vividly illustrate the system's capacity to comprehend and reason contents across various modalities in any combination. Please kindly refer to Appendix §J for the demonstrations.

## 7 CONCLUSION

In this work, we presented an end-to-end general-purpose any-to-any multimodal Large Language Model (MM-LLM). By connecting an LLM with multimodal adaptors and different diffusion decoders, NExT-GPT is capable of perceiving inputs and generating outputs in any combination of text, image, video, and audio. Harnessing the existing well-trained highly-performing encoders and decoders, training NExT-GPT only entails a few number of parameters (1%) of certain projection layers, which not only benefits low costs but also facilitates convenient expansion of more potential modalities in the future. To enable our NExT-GPT with complex cross-modal semantic understanding and content generation, we further introduced a modality-switching instruction tuning (MosIT), and manually curated a high-quality dataset for MosIT. Overall, our research showcases the potential of any-to-any MM-LLMs in bridging the gap between various modalities and paving the way for more human-like AI systems in the future.

ETHICS STATEMENT

Here we discuss the primary ethical considerations of the NExT-GPT model and also the `MosIT` dataset.

**Use of Generative Content**    The NExT-GPT, limited by the quantity of fine-tuning data and the quality of the base models, may generate some low-quality content. Also, as a generative model, the LLM will produce hallucinated content in multimodal formats that may be harmful to society. We have reminded users to interpret the results with caution. Anyone who uses this LLM should obey the rules in a license. And also commercial use of our system is not allowed.

**Intellectual Property Protection**    Concerning some multimodal contents, i.e., image, video, and audio, are collected from social media platforms, such as Youtube, and Twitter, we uphold the importance of data privacy and ensure that all data collection adheres to the terms and conditions of the respective social media platforms. Where applicable, we seek and obtain consent from users or content creators before including their data in our dataset.

**Privacy Claim**    We take meticulous care to anonymize and protect the identities of individuals and organizations mentioned in the dataset. Any personally identifiable information is removed or obfuscated to safeguard privacy.

**Bias Mitigation.**    We remain vigilant in minimizing bias in dataset collection, striving to ensure that our dataset is representative and does not disproportionately favor or disfavor any particular group or perspective.

**Research Integrity**    We pledge to employ the dataset for research and analysis purposes that uphold the highest standards of integrity, without engaging in activities that could harm individuals or organizations mentioned in the dataset.

**Continuous Monitoring and Improvement**    We commit to continuously monitor and assess our dataset collection practices to identify and rectify any ethical issues that may arise. We also welcome feedback from the community to enhance the ethical aspects of our work.

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

## A    LIMITATION AND FUTURE WORK

As future work, there are at least following four avenues to explore.

- **i) Modalities & Tasks Expansion**: Due to resource limitations, currently, our system supports input and output in four modalities: language, images, videos, and audio. Next, we plan to extend this to accommodate even more modalities (e.g., web page, 3D vision, heat map, tables&figures) and tasks (e.g., object detection, segmentation, grounding, and tracking), broadening the system's applicability to become more universal.
- **ii) LLM Variants**: Currently, we have implemented the 7B Vicuna version of the LLM. Our next plans involve incorporating various LLM types and sizes, allowing practitioners to choose the most suitable one for their specific requirements.
- **iii) Multimodal Generation Strategies**: While our system excels in generating content across modalities, the quality of generative outputs can sometimes be limited by the capabilities of the diffusion model. It is very promising to explore the integration of retrieval-based approaches to complement the generative process, potentially improving the overall system's performance.
- **iv) MosIT Dataset Expansion**: Currently, our IT dataset has room for expansion. We intend to significantly increase the amount of annotated data, ensuring a more comprehensive and diverse set of instructions to further enhance the MM-LLMs' ability to understand and follow user prompts effectively.

## B    FULL RELATED WORK

**Cross-modal Understanding and Generation**    Our world is replete with multimodal information, wherein we continuously engage in the intricate task of comprehending and producing cross-modal content. The AI community correspondingly emerges varied forms of cross-modal learning tasks, such as Image/Video Captioning (Zeng et al., 2023; Dessì et al., 2023; Milewski et al., 2020;?; Gu et al., 2023; Lin et al., 2022), Image/Video Question Answering (Yang et al., 2021; Xiao et al., 2022; Li et al., 2022; Yu et al., 2023; Anderson et al., 2018), Text-to-Image/Video/Speech Synthesis (Singer et al., 2022; Hong et al., 2022; Voynov et al., 2023; Gal et al., 2022; Ding et al., 2021; Liu et al., 2023a; Huang et al., 2023a), Image-to-Video Synthesis (Dorkenwald et al., 2021; Karras et al., 2023) and more, all of which have experienced rapid advancements in past decades. Researchers have proposed highly effective multimodal encoders, with the aim of constructing unified representations encompassing various modalities. Meanwhile, owing to the distinct feature spaces of different modalities, it is essential to undertake modality alignment learning. Moreover, to generate high-quality content, a multitude of strong-performing methods have been proposed, such as Transformer (Vaswani et al., 2017; Zhang et al., 2022; Ding et al., 2021; Ge et al., 2022), GANs (Liu et al., 2020; Brock et al., 2019; Xu et al., 2018; Zhu et al., 2019), VAEs (Vahdat & Kautz, 2020; Razavi et al., 2019), Flow models (Shibata et al., 2022; Bashiri et al., 2021) and the current state-of-the-art diffusion models (Hoogeboom et al., 2021; Qu et al., 2023b; Mou et al., 2023; Feng et al., 2022; Rombach et al., 2022). Especially, the diffusion-based methods have recently delivered a remarkable performance in a plethora of cross-modal generation tasks, such as DALL-E (Ramesh et al., 2021), Stable Diffusion (Rombach et al., 2022). While all previous efforts of cross-modal learning are limited to the comprehension of multimodal inputs only, CoDi (Tang et al., 2023) lately presents groundbreaking development. Leveraging the power of diffusion models, CoDi possesses the ability to generate any combination of output modalities, including language, images, videos, or audio, from any combination of input modalities in parallel. Regrettably, CoDi might still fall short of achieving human-like deep reasoning of input content, with only parallel cross-modal feeding&generation.

**Multimodal Large Language Models**    LLMs have already made profound impacts and revolutions on the entire AI community and beyond. The most notable LLMs, i.e., OpenAI's ChatGPT (OpenAI, 2022a) and GPT4 (OpenAI, 2022b), with alignment techniques such as instruction tuning (Ouyang et al., 2022; Li et al., 2023f; Zhang et al., 2023d; Liu et al., 2023b) and reinforcement learning from human feedback (RLHF) (Stiennon et al., 2020), have demonstrated remarkable language understanding and reasoning abilities. And a series of open-source LLMs, e.g., Flan-T5 (Chung et al., 2022), Vicuna (Chiang et al., 2023), LLaMA (Touvron et al., 2023) and Alpaca (Taori et al., 2023), have greatly spurred advancement and made contributions to the community (Zhu et al., 2023; Zhang et al., 2023a). Afterward, significant efforts have been made to construct LLMs dealing with multimodal inputs and tasks, leading to the development of MM-LLMs.

On the one hand, most of the researchers build fundamental MM-LLMs by aligning the well-trained encoders of various modalities to the textual feature space of LLMs, so as to let LLMs perceive other modal inputs (Huang et al., 2023c; Zhu et al., 2023; Su et al., 2022; Koh et al., 2023). For example, Flamingo (Alayrac et al., 2022) uses a cross-attention layer to connect a frozen image encoder to the LLMs. BLIP-2 (Li et al., 2023c) employs a Q-Former to translate the input image queries to the LLMs. LLaVA (Liu et al., 2023b) employs a simple projection scheme to connect image features into the word embedding space. There are also various similar practices for building MM-LLMs that are able to understand videos (e.g., Video-Chat (Li et al., 2023d) and Video-LLaMA (Zhang et al., 2023c)), audios (e.g., SpeechGPT (Zhang et al., 2023b)), etc. Profoundly, PandaGPT (Su et al., 2023) achieves a comprehensive understanding of six different modalities simultaneously by integrating the multimodal encoder, i.e., ImageBind (Girdhar et al., 2023).

Nevertheless, these MM-LLMs all are subject to the limitation of only perceiving multimodal data, without generating content in arbitrary modalities. To achieve LLMs with both multimodal input and output, some thus explore employing LLMs as decision-makers, and utilizing existing off-the-shelf multimodal encoders and decoders as tools to execute multimodal input and output, such as Visual-ChatGPT (Wu et al., 2023), HuggingGPT (Shen et al., 2023), and AudioGPT (Huang et al., 2023b). As aforementioned, passing messages between modules with pure texts (i.e., LLM textual instruction) under the discrete pipeline scheme will inevitably introduce noises. Also lacking comprehensive tuning across the whole system significantly limits the efficacy of semantics understanding. Our work takes the mutual benefits of both the above two types, i.e., learning an any-to-any MM-LLM in an end-to-end manner.

## C   MODEL TRAINING

For NExT-GPT model training, we consider a three-stage learning process:

- **Stage-1: Encoding-size Alignment Learning**. The input projection layer is one linear layer with a hidden size of 4096. As discussed in Section §4.1, we bridge the alignment to perform the caption generation task. The cross-entropy is employed as the loss function. During training, we only keep the input projection layer trainable while the other part of NExT-GPT is frozen. We employ Adam (Kingma & Ba, 2015) optimizer to update the parameters. This stage can be understood as training a compatible multimodal tokenizer for the frozen LLM.
- **Stage-2: Decoding-side Alignment Learning**. The output projection layer adopts a transformer-based architecture characterized by a hidden size of 512, 4 attention heads, 4 encoder layers, and 4 decoder layers. Additionally, the dropout ratio is set as 0.1. The optimization process for the three output projection layers involves a combination of two training objectives: cross-entropy focusing on the generated signal tokens, and $l_2$-distance measuring the alignment between the representation of signal tokens and captions, as shown in Section §4.2. We employ the Adam optimizer for this stage, with only the parameters of the output projection layers being learnable, while others remain frozen.
- **Stage-3: End-to-end Instruction-Tuning**. In this stage. we train the whole NExT-GPT using instruction-following datasets, as enumerated in Section §5.2. We incorporate LoRA to fine-tune the weights of the LLM. Moreover, both the input and output projection layers are trainable. The training objectives include two parts: 1) cross-entropy between the generated and gold response, 2) $l_2$-distance between the representation of signal tokens and captions. The Adam optimizer is applied to update the learnable parameters.

## D   DETAILED DATASET

Here, we enumerate the datasets employed for training and evaluating NExT-GPT:

- **'Text-X' Pair Dataset**.
  - **CC3M** (Sharma et al., 2018): contains over 3 million images accompanied by diverse styles of natural-language descriptions.
  - **COCO-caption** (Lin et al., 2014): is a large-scale image-text pair dataset which is taken as image captioning, or text-to-image generation task benchmark.
  - **WebVid-2M** (Bain et al., 2021): is a large-scale dataset of short videos with textual description sourced from stock footage sites.

- **MSR-VTT** (Xu et al., 2016): is a large-scale dataset for the open domain video captioning, which consists of 10,000 video clips from 20 categories, and each video clip is annotated with 20 English sentences by Amazon Mechanical Turks.
- **AudioCaps** (Kim et al., 2019): with 46K audio-text pairs derived from the AudioSet (Gemmeke et al., 2017) dataset.
- **Text-conditioned 'X' Editing Dataset**.
  - **VCTK** (Veaux et al., 2017): includes speech data uttered by 110 English speakers with various accents. Each speaker reads out about 400 sentences, which were selected from a newspaper, the rainbow passage and an elicitation paragraph used for the speech accent archive.
  - **DAVIS** (Perazzi et al., 2016): comprises a total of 50 sequences, 3455 annotated frames, all captured at 24fps and Full HD 1080p spatial resolution. The editing prompts of the videos are collected or generated by Ceylan et al. (2023).

## E  INFERENCE PROCESS

In Figure 4 we further illustrate the inference procedure of NExT-GPT. Given certain user inputs of any combination of modalities, the corresponding modal encoders, and projectors transform them into feature representations and pass them to LLM[7]. Then, LLM decides what content to generate, i.e., textual tokens, and modality signal tokens. If LLM identifies a certain modality content (except language) to be produced, a special type of token (Koh et al., 2023) will be output indicating the activation of that modality; otherwise, no special token output means deactivation of that modality.

## F  HUMAN EVALUATION ON COMPLEX ANY-TO-ANY QA

We also carry out evaluation on some more scenarios where there are complicated cross-modal interactions between inputs and outputs. We mainly compare the model performance for the settings with different modality conversions. As no standard benchmark can be leveraged, here we adopt human evaluation. We ask several evaluators to score the performance of NExT-GPT on a scale from 1 to 10. Figure 5 shows the comparisons. We find NExT-GPT is more competent in producing images, compared with the generations on videos and audio. Also generating mixed combinations of multimodal content is slightly inferior to the generation of single-modal content, due to the complexity of the latter.

## G  HUMAN EVALUATION ON PIPELINE VS END-TO-END MM-LLMS

Intuitively, LLM outputs textual captions and feeds to the follow-up diffusion models for generation, which is one type of prior existing method for reaching the goal of unified MM-LLM systems. Here, we conducted experiments to compare with such a pipeline-style baseline. We consider two types of comparisons: 1) Visual-ChatGPT (Wu et al., 2023) and HuggingGPT (Shen et al., 2023), which are existing systems that have free open access; 2) NExT-GPT variant with captions as the messenger (which we mark as NExT-GPT-caption). To implement NExT-GPT-caption, the captions directly generated by LLM will be fed into the following generation models, instead of using the soft representations of the signal tokens. As Visual-ChatGPT only supports image generation, we here consider the evaluation on the Text-to-Text&Image setting.

To evaluate if the system really or how well understands the input and generates output content (response text + image), we perform the human evaluation. For constructing the testing data, we first leverage GPT-4 to synthesize 1) 100 **simple** instructions (e.g., involving short and simple semantic content) that can explicitly prompt MM-LLMs to generate images, and 2) 100 **complex** instructions (e.g., involving intricate and semantically-rich scenes) that require implicit reasoning ability to generate image content. Then, the synthesized instructions are fed into the models to generate the response text + image content. Subsequently, five unbiased volunteers evaluate the generated results under three aspects, 1) **Instruction following**, identifying, among the four models, which of the generated text+image accurately responded to the input instructions, 2) **Rationality**, determining

---

[7]Except the text inputs, which will be directly fed into LLM.

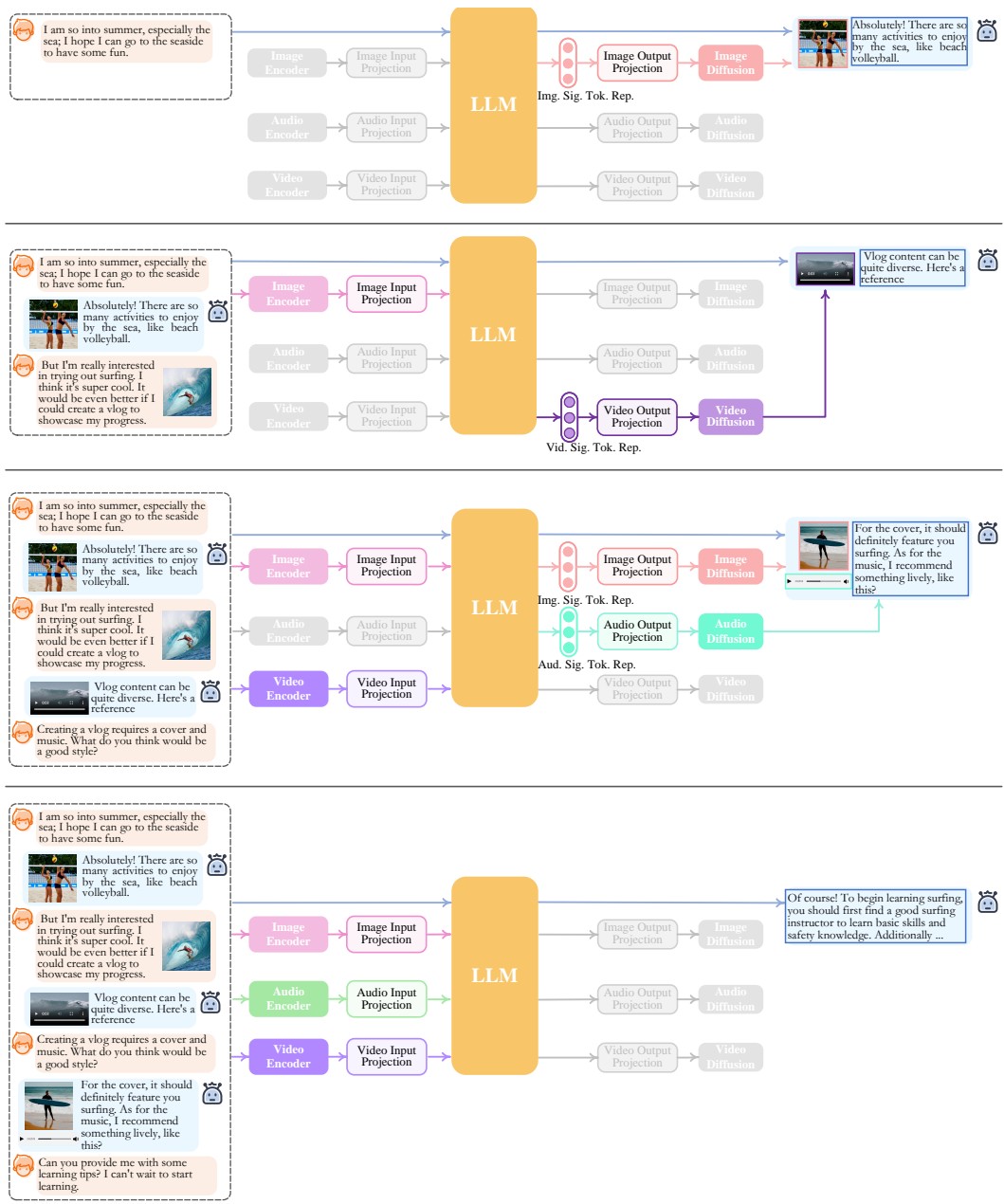

Figure 4: NExT-GPT inference process. Grey colors denote the deactivation of the modules.

which of the generated images adhered to the input instructions, 3) **Quality**, evaluating which of the generated images exhibited the highest quality.

The evaluation results are shown in Table 14, where we can notice the interesting observation. On the simple instructions (first three columns), mostly these four models perform at similar levels. This means the impacts could be quite limited whether we take a pipeline modeling or end-to-end system on the comparatively simple user inputs. But on complex instructions, ours performs significantly better than two existing systems and NExT-GPT-caption in terms of the instruction-following capability and image generation quality. Notably, a notable degradation in the quality of generated images is observed when captions are utilized as messengers compared to the instruction-following performances. This highlights the inadequacy of captions in conveying the necessary information for generating complex images.

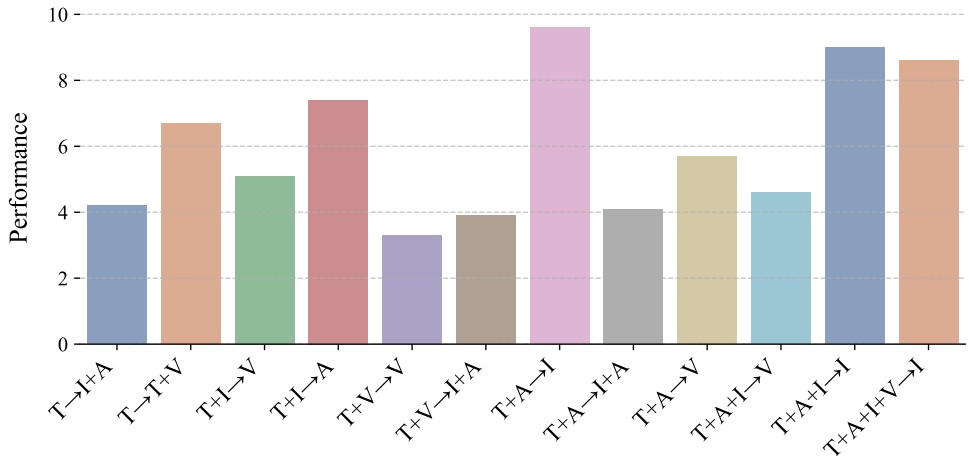

Figure 5: Comparative performance of NExT-GPT on various complex cross-modal conversions.

| Model | Simple Instruction | | | Complex Instruction | | |
|---|---|---|---|---|---|---|
| | Instruction Following | Rationality | Quality | Instruction Following | Rationality | Quality |
| HuggingGPT | 94 | 92 | 87 | 82 | 74 | 73 |
| Visual-ChatGPT | 94 | 90 | 86 | 84 | 76 | 72 |
| NExT-GPT-caption | 93 | 90 | 81 | 76 | 68 | 65 |
| NExT-GPT | 95 | 92 | 89 | 84 | 83 | 80 |

Table 14: Human Evaluation (1-100 scale, results are on average) of NExT-GPT in comparison with pipeline baselines that directly generate captions for downstream generation models.

## H  CASE STUDY ON PIPELINE-STYLE VS. END-TO-END UNIFICATION

We earlier have elaborated on the difference as well as the necessity of building a unified any-to-any multimodal LLM in an end-to-end manner, compared with the existing pipeline-style systems that generate intermediate captions and then pass to the downstream tools (e.g., diffusion models for generation). The cascade process inevitably introduces noise and propagates errors. Meanwhile, the entire system only leverages existing pre-trained tools for inference only, whereas without an end-to-end updating throughout the whole system, the capability in more accurately interpreting complex user instructions and generating content will be compromised. Here we add few illustrations, where we make comparisons with these pipeline-style systems: 1) Visual-ChatGPT and HuggingGPT, which are existing systems that have free open access; 2) NExT-GPT variant with captions as the messenger (which we mark as NExT-GPT-caption). To implement NExT-GPT-caption, the captions directly generated by LLM will be fed into the following generation models, instead of using the soft representations of the signal tokens. As Visual-ChatGPT only supports image generation, we here consider the evaluation on the Text-to-Text&Image setting.

Figure 6 presents the case of image generation from a simple input user instruction; while Figure 7 and 8 present two cases of image generation from comparatively complex input user instructions. On the simple one, all generated image content from both pipeline-style and end-to-end (ours) systems seem correct and coincide with the input prompt. However, when handling the complex instructions, as seen in Figure 7 and 8, the generated image content can be wrong and biased to the user intention. The problems are rooted in the core of different modalities, i.e., there are inherent gaps between language and visual modalities that cannot be eliminated. Here are two representative attributes: **the numeration of vision** (cf. Figure 7) and **the visual-spatial relational semantics** (cf. Figure 8), which could be hard to (or even cannot) be expressed by the intermediate captions conveniently. Utilizing textual captions as intermediate representations runs the risk of overlooking these modality-specific features when expressing non-linguistic (e.g., visual) modalities solely through language.

By the way, we kindly note a fact that, with the intermediate captions produced from the pipeline-style systems in Figure 7 and 8, the Stable Diffusion model just has difficulty in accurately understanding the vision numeration and visual-spatial relation and generating correct answers, i.e., they are the problems inherent to the Stable Diffusion model itself, and Stable Diffusion alone is tricky to

**Instruction (a):** Generate me a figure of a cat running in the garden.

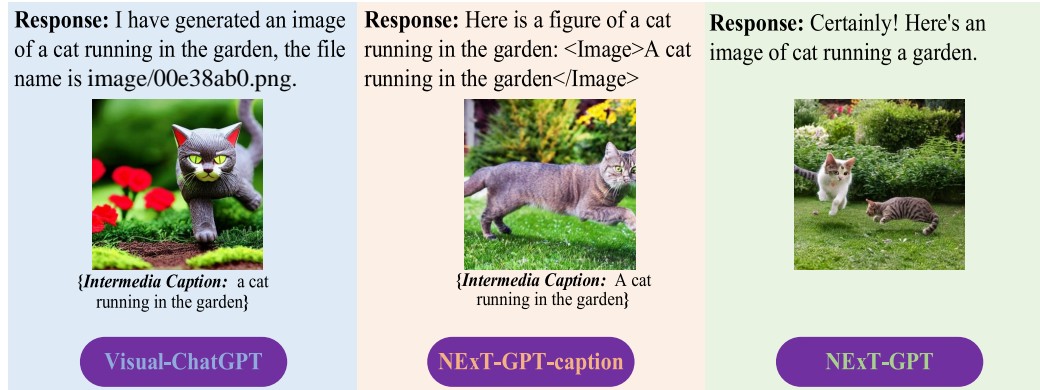

**Response:** I have generated an image of a cat running in the garden, the file name is image/00e38ab0.png.

{*Intermedia Caption:* a cat running in the garden}

**Visual-ChatGPT**

**Response:** Here is a figure of a cat running in the garden: <Image>A cat running in the garden</Image>

{*Intermedia Caption:* A cat running in the garden}

**NExT-GPT-caption**

**Response:** Certainly! Here's an image of cat running a garden.

**NExT-GPT**

Figure 6: Illustration of case study, image generation from a simple instruction on Visual-ChatGPT, NExT-GPT-caption, and NExT-GPT.

**Instruction (b):** There were originally 7 apples on the table, but I ate one. Then, how many apples are left now? Please generate a picture to describe the result.

**Response:** Sure, I'd be happy to help you understand the process and results.
Based on your statement, "There were originally 7 apples on the table, but I ate one", I inferred that there are now 6 apples left on the table.
To visualize this, I used a text-to-image model to generate an image based on your description. The model I used is called 'runwayml/stable-diffusion-v1-5', which is currently the only available model for this task. The image generated can be found at the following path: '/images/ec4d.jpg'.
Please note that the image is a representation of the text and may not be 100% accurate. If there's anything else you need, feel free to ask!

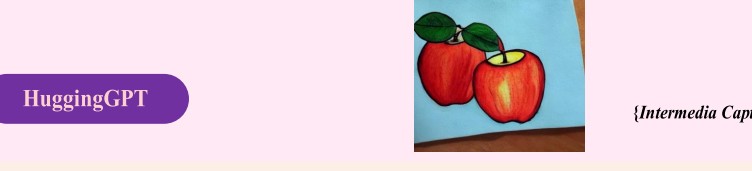

**HuggingGPT**

{*Intermedia Caption:* 6 apples left on the table}

**Response:** Sure! As you can see, the number of apples has decreased from 7 to 6. Here's an image that shows the results after one apple was eaten: <Image>An image of a table with 6 apples</Image>

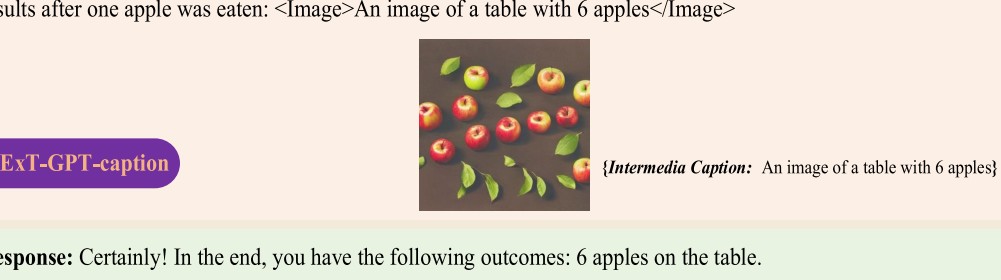

**NExT-GPT-caption**

{*Intermedia Caption:* An image of a table with 6 apples}

**Response:** Certainly! In the end, you have the following outcomes: 6 apples on the table.



**NExT-GPT**

Figure 7: Illustration of image generation from a complex instruction on HuggingGPT, NExT-GPT-caption, and NExT-GPT. In this case, the numeration understanding of vision has been wrong by pipeline-style methods.

**Instruction (c):** In the park, an old lady is sitting to the right of an old man, and a small dog with yellow and white fur is lying at the old man's arm. What a heartwarming scene! Can you help me visualize it in an image?

**Response:** Sure, I'd be happy to help you visualize the scene you described. Based on your description, I used a model to generate an image that represents the scene. Unfortunately, as an AI, I can't directly show you the image, but I can tell you where to find it. The generated image is stored at this location: '/images/a62c.jpg'.
To accomplish this, I used the 'runwayml/stable-diffusion-v1-5' model. This model was chosen because it was the only one available for this task. The model took your description and used it to generate an image that represents the scene you described.
Please note that while the model does its best to accurately represent the scene, the results may not always be perfect due to the complexity of interpreting and visualizing text descriptions.

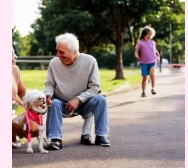

**HuggingGPT**

{*Intermedia Caption:* In the park, an old lady is sitting to the right of an old man, and a small dog with yellow and white fur is lying at the old man's arms}

**Response:** Certainly! Here's an image that captures an old woman is sitting on a bench, surrounded by a park with a small dog lying at the old man's arms. <Image>An image of an old woman sitting on a bench, surrounded by a park with a small dog lying at the old man's arms</Image>.

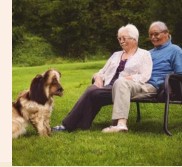

**NExT-GPT-caption**

{*Intermedia Caption:* An image of an old woman sitting on a bench, surrounded by a park with a small dog lying at the old man's arms}

**Response:** Certainly! Here's an image that captures the warmth and companionship between the old lady and the old man, as well as the peaceful presence of the small dog at their arms.

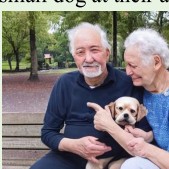

**NExT-GPT**

Figure 8: Illustration of image generation from another complex instruction on HuggingGPT, NExT-GPT-caption, and NExT-GPT. In this case, the understanding of visual-spatial relational semantics has been wrong by pipeline-style methods.

overcome. Most recent work tries to solve this issue by integrating the vision-specific features into the Stable Diffusion (Feng et al., 2023; Qu et al., 2023a) via additional feature engineering. But, in our NExT-GPT with an end-to-end solution, the implicit modality signal token embeddings that carry rich modality-specific features of non-linguistic will be naturally encoded and passed to the downstream modules (e.g., Stable Diffusion), without any further external effort.

## I MULTIMODAL IT DATASETS COMPARISON

Here, we compare the existing multimodal instruction tuning (IT) datasets, as detailed in Table 15. As can be seen, the response modality of the existing IT datasets is merely limited to text. In this work, we leverage GPT-4 to generate a T2M IT dataset, comprising 15k instances, which serves as a foundation for instructing the model to generate responses in other modalities, such as image, video, and audio. Furthermore, we construct a modality-switching IT dataset with 5k instances, named `MosIT`. This dataset is designed to emulate the human-machine complex interaction featuring diverse and dynamic shifts in modalities within both inputs and outputs.

| Dataset | Data Source | In→Out Modality | Approach | Multi-turn Reason | #Img/Vid/Aud | #Dialog Turn. | #Instance |
|---|---|---|---|---|---|---|---|
| **▶ Existing data** | | | | | | | |
| MiniGPT-4 (Zhu et al., 2023) | CC , CC3M | T+I→T | Auto | ✗ | 134M/-/- | 1 | 5K |
| StableLLaVA (Li et al., 2023f) | SD | T+I→T | Auto+Manu. | ✗ | 126K/-/- | 1 | 126K |
| LLaVA (Zhang et al., 2023d) | COCO | T+I→T | Auto | ✓ | 81K/-/- | 2.29 | 150K |
| SVIT (Zhao et al., 2023a) | MS-COCO, VG | T+I→T | Auto | ✓ | 108K/-/- | 5 | 3.2M |
| LLaVAR (Zhang et al., 2023d) | COCO , CC3M , LAION | T+I→T | LLaVA+Auto | ✓ | 20K/-/- | 2.27 | 174K |
| VideoChat (Li et al., 2023d) | WebVid | T+V→T | Auto | ✓ | -/8K/- | 1.82 | 11K |
| Video-ChatGPT (Maaz et al., 2023) | ActivityNet | T+V→T | Inherit | ✗ | -/100K/- | 1 | 100K |
| Video-LLaMA (Zhang et al., 2023c) | MiniGPT-4, LLaVA, VideoChat | T+I/V→T | Auto | ✓ | 81K/8K/- | 2.22 | 171K |
| InstructBLIP (Dai et al., 2023) | Multiple | T+I/V→T | Auto | ✗ | - | - | ~1.6M |
| MIMIC-IT (Li et al., 2023a) | Multiple | T+I/V→T | Auto | ✗ | 8.1M/502K/- | 1 | 2.8M |
| PandaGPT (Su et al., 2023) | MiniGPT-4, LLaVA | T+I→T | Inherit | ✓ | 81K/-/- | 2.29 | 160K |
| MGVLID (Zhao et al., 2023b) | Multiple | T+I+B→T | Auto+Manu. | ✗ | 108K/-/- | - | 108K |
| M³IT (Li et al., 2023e) | Multiple | T+I/V/B→T | Auto+Manu. | ✗ | -/-/- | 1 | 2.4M |
| LAMM (Yin et al., 2023) | Multiple | T+I+PC→T | Auto+Manu. | ✓ | 91K/-/- | 3.27 | 196k |
| BuboGPT (Zhao et al., 2023c) | Clotho, VGGSS | T+A/(I+A)→T | Auto | ✗ | 5k/-/9K | - | 9K |
| mPLUG-DocOwl (Ye et al., 2023a) | Multiple | T+I/Tab/Web→T | Inherit | ✗ | - | - | - |
| **▶ In this work** | | | | | | | |
| T2M | Webvid, CC3M, AudioCap | T→T+I/A/V | Auto | ✗ | 5K/5K/5K | 1 | 15K |
| MosIT | Youtube, Google, Flickr, Midjourney, etc. | T+I+A+V→T+I+A+V | Auto+Manu. | ✓ | 4K/4K/4K | 4.8 | 5K |

Table 15: Summary and comparison of existing datasets for multimodal instruction tuning. T: text, I: image, V: video, A: audio, B: bounding box, PC: point cloud, Tab: table, Web: web page.

## J EXAMPLE DEMONSTRATIONS

Figure 9, 10, 11, 12, 13 and 14 show several real examples produced by NExT-GPT.

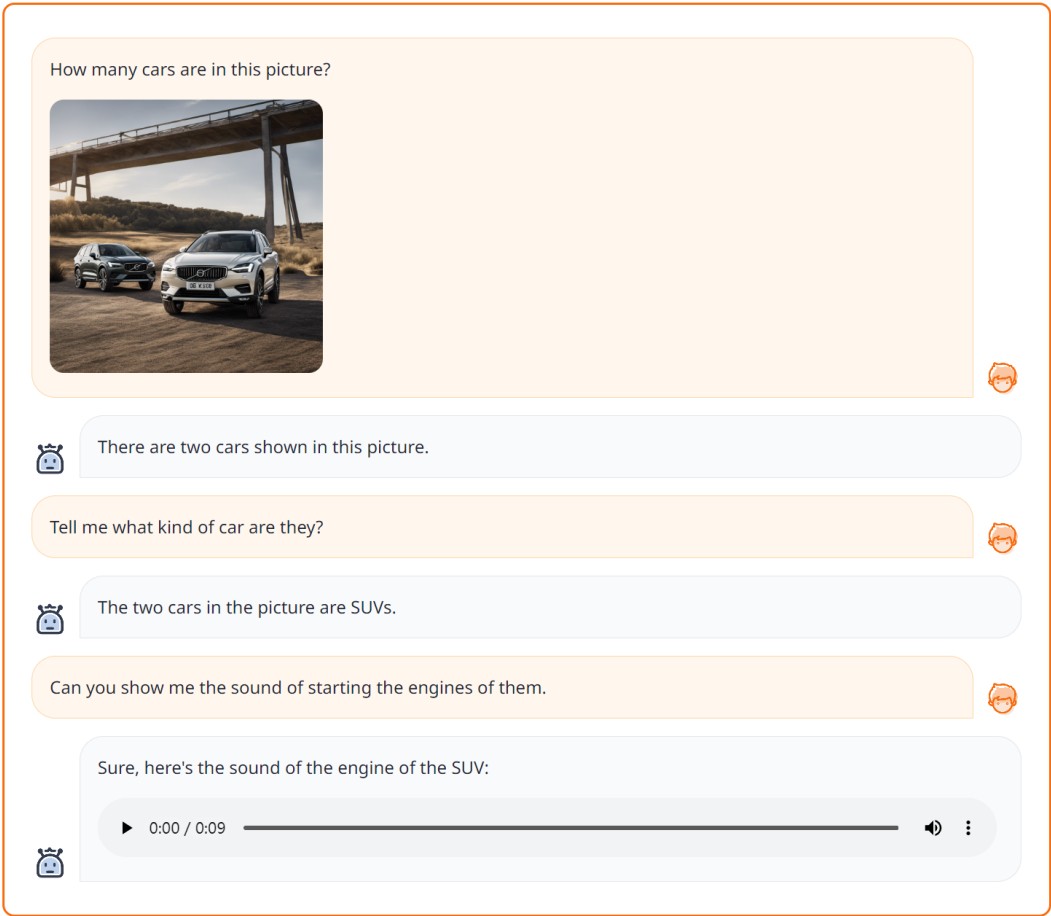

Figure 9: Example of Text+Image → Text+Audio.

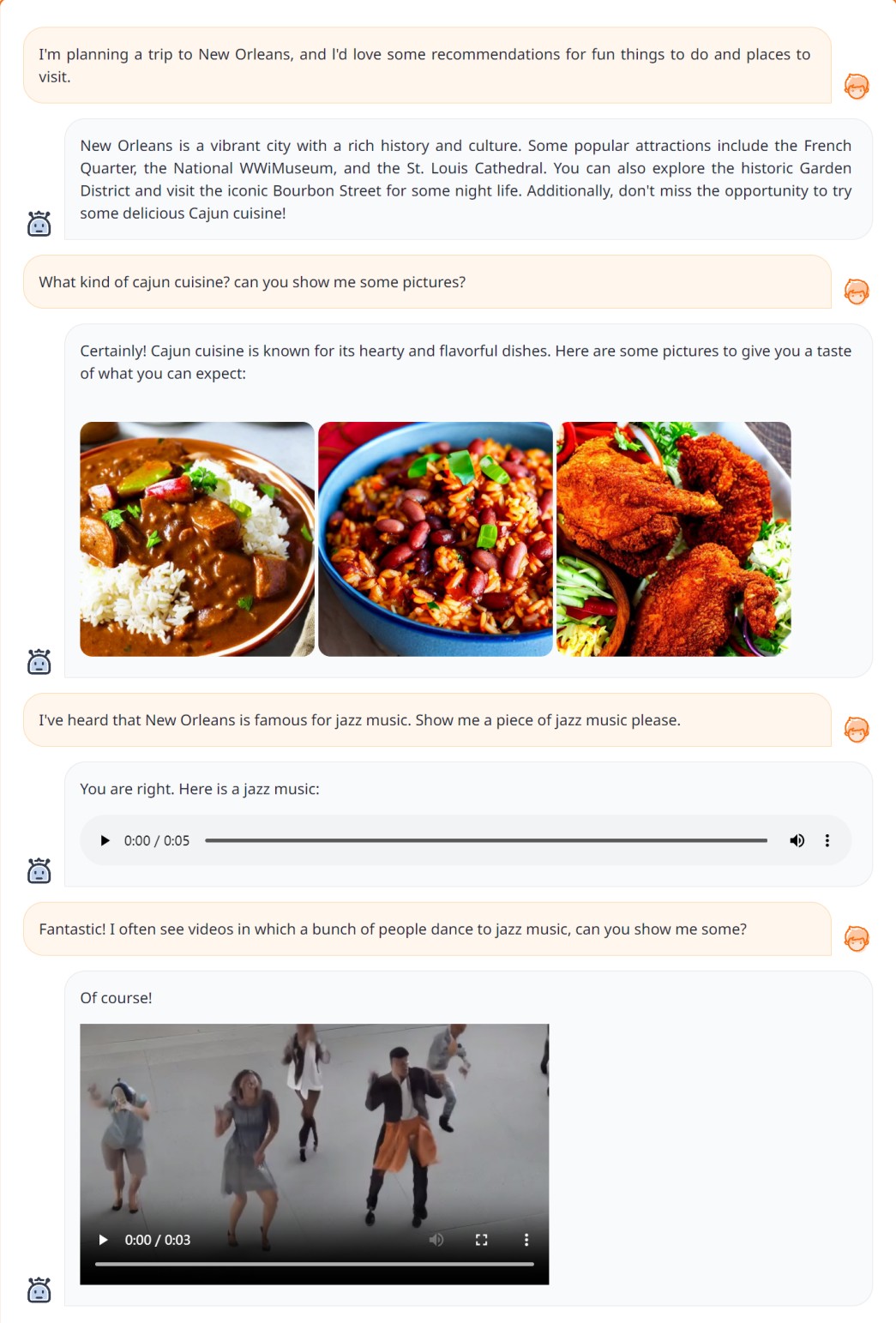

Figure 10: Example of Text → Text+Image+Video+Audio.

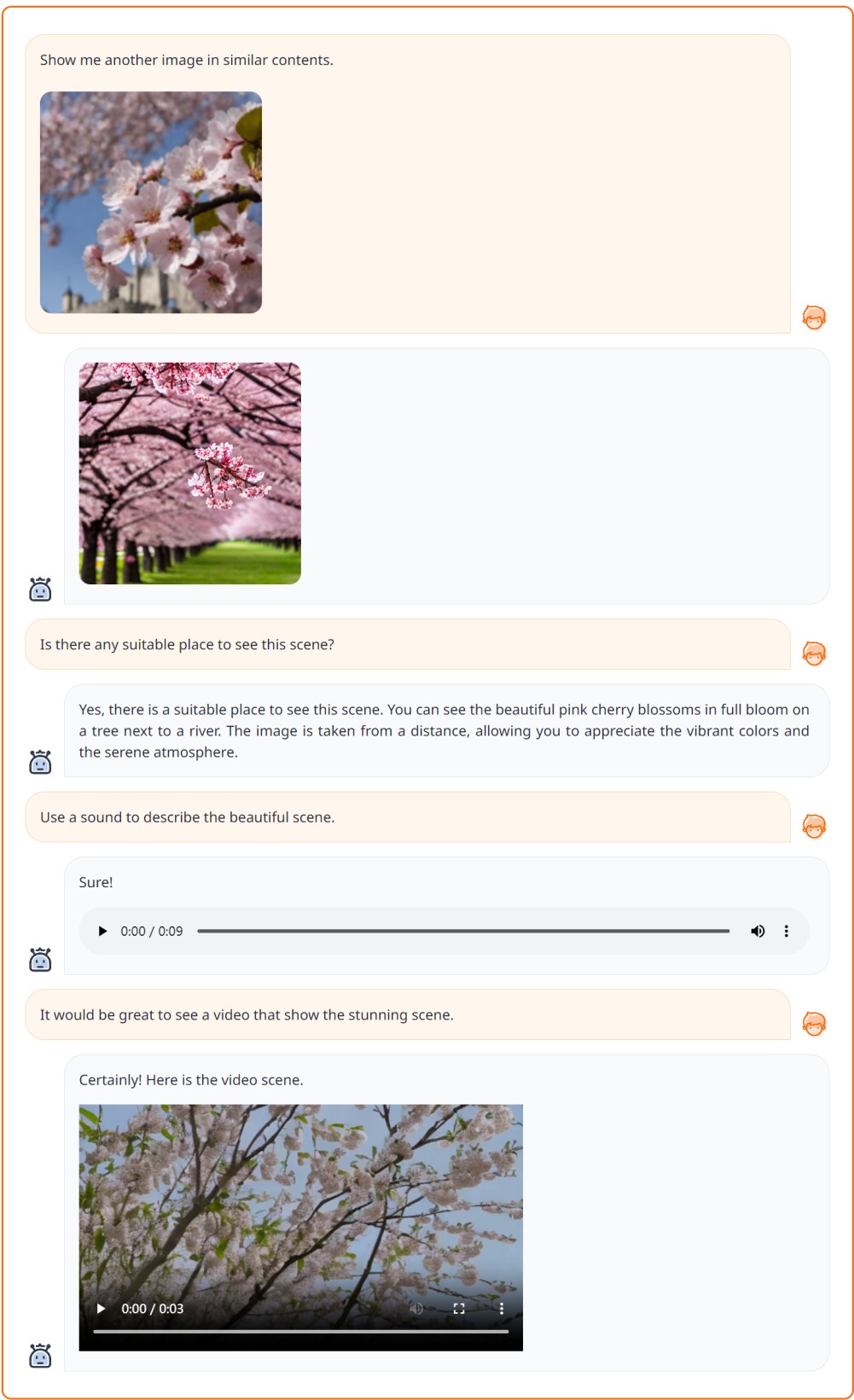

Figure 11: Example of Text+Image → Text+Image+Video+Audio.

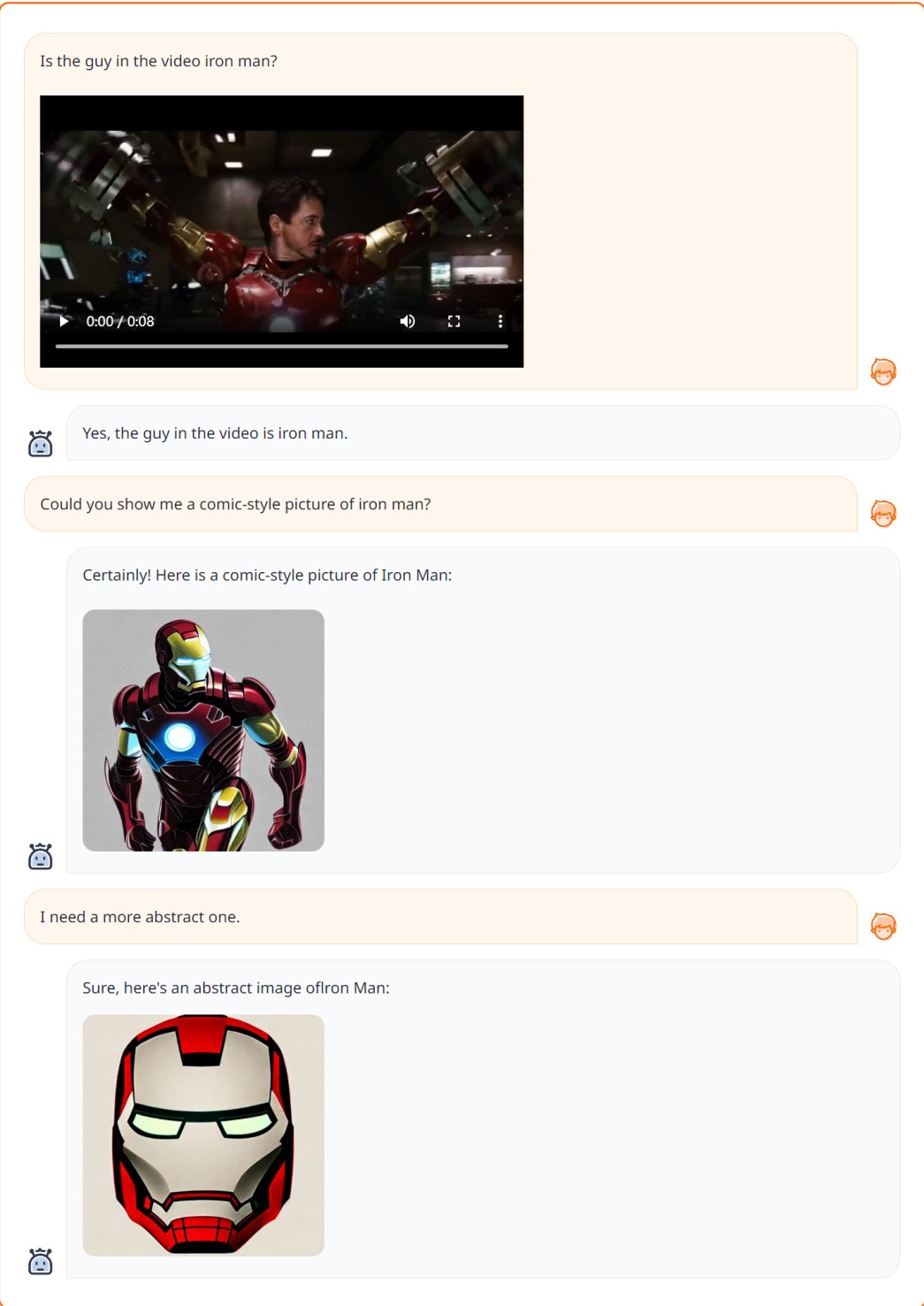

Figure 12: Example of Text+Video → Text+Image.

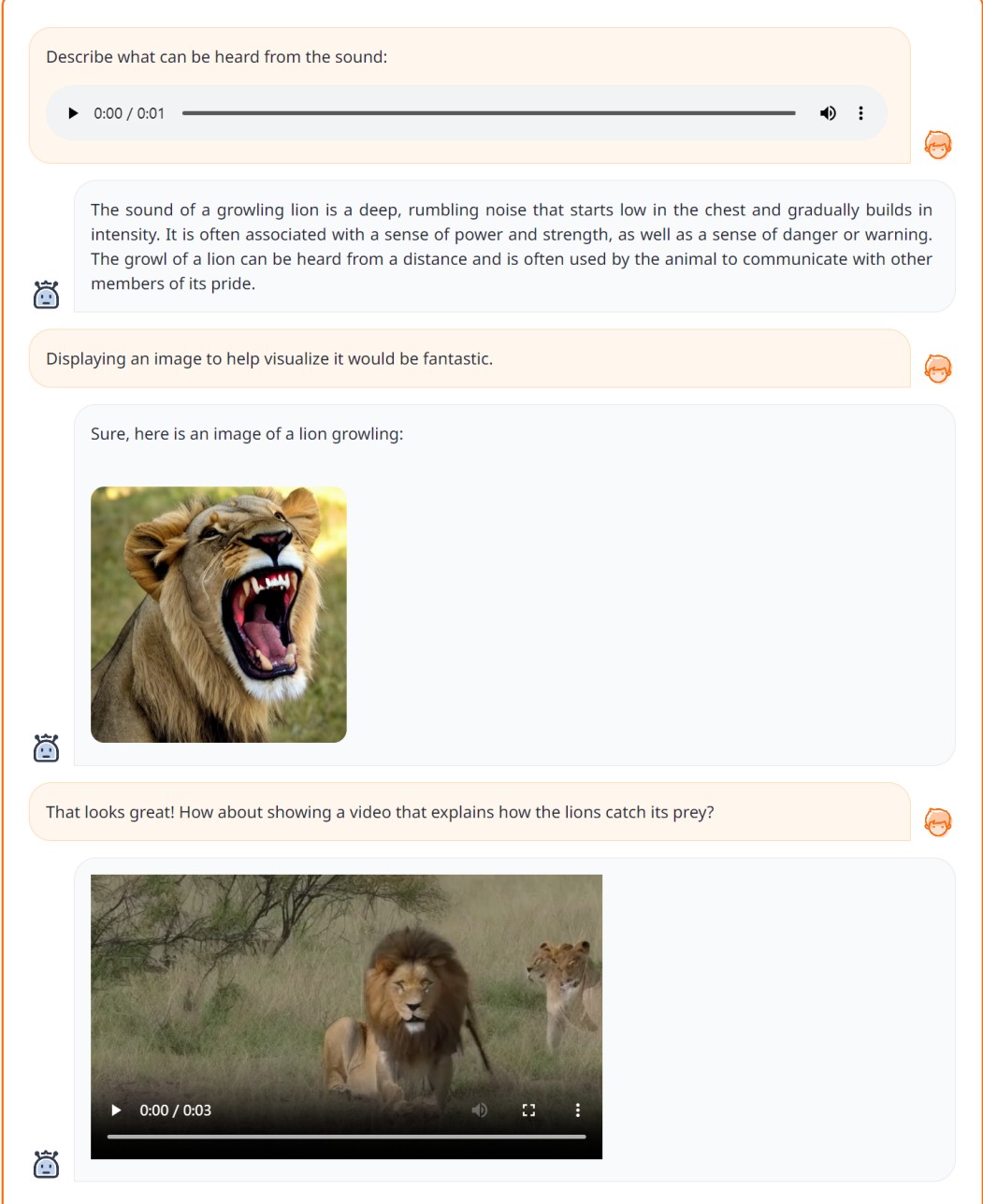

Figure 13: Example of Text+Audio → Text+Image+Video.

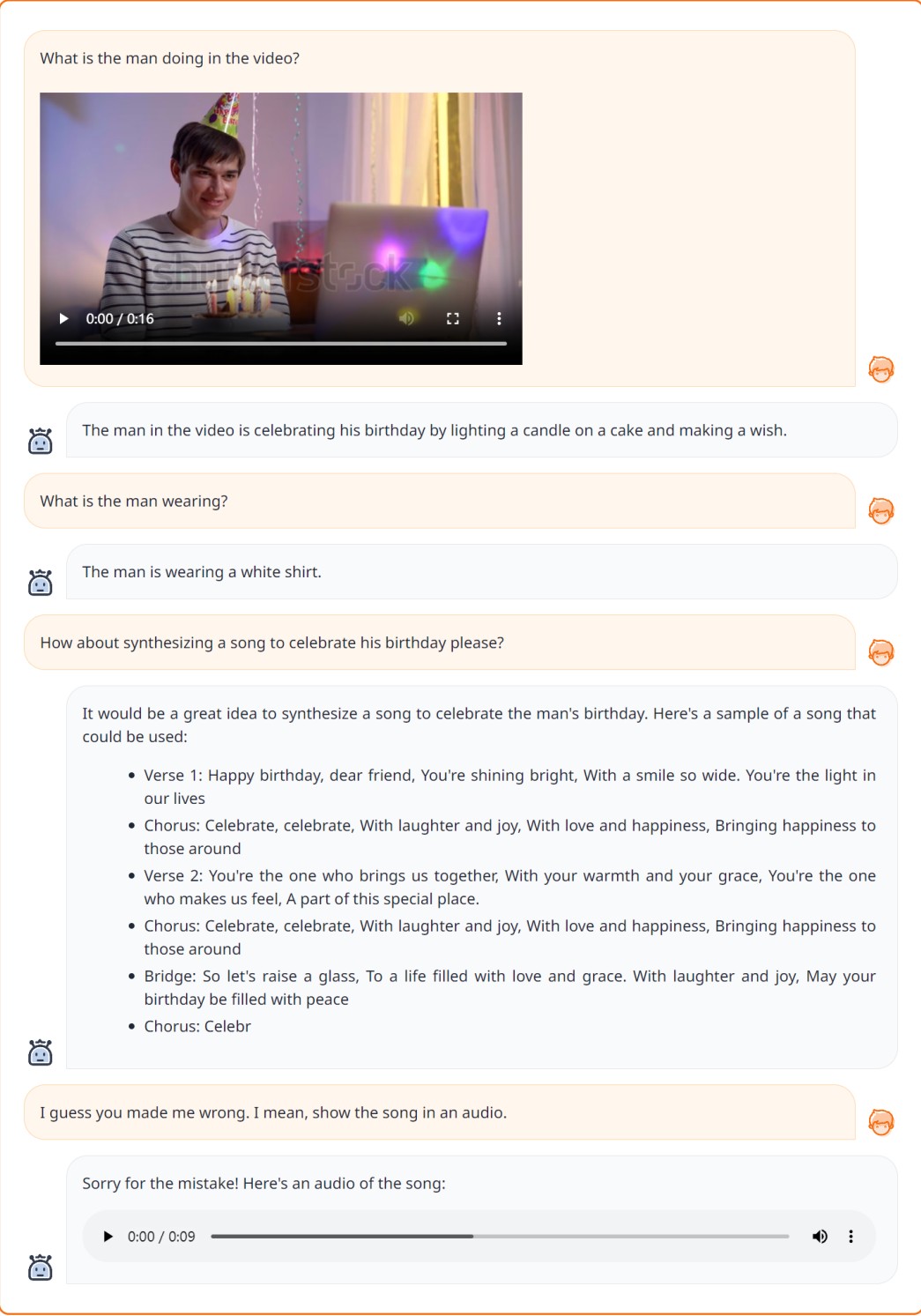

Figure 14: Example of Text+Video → Text+Audio.

