# OpenReview forum: "NExT-GPT: Any-to-Any Multimodal LLM"
_ICLR.cc/2024/Conference — Submitted to ICLR 2024_

### Official Review · Reviewer_mDkx · 2023-10-27

**Soundness:** 3 good
**Presentation:** 3 good
**Contribution:** 3 good
**Rating:** 8
**Confidence:** 4

**Summary:**

This paper proposed an end-to-end general-purpose any-to-any MM-LLM system, NExT-GPT, by connecting an LLM with multimodal adaptors and different diffusion decoders.

**Strengths:**

(1) The paper formulation is good and clear.
(2) Introduced lightweight alignment learning techniques.
(3) Annotated a high-quality modality-switching instruction tuning dataset.

**Weaknesses:**

(1) The model relies on different pretrained models to understand different types of information, like text, images, and audio. The quality of pretraining may directly impact how well the model performs its tasks.
(2) What is the size of parameters when tuning the model for each modality?

**Questions:**

(1) Please see the comments above.
(2) The work in [1] may be related, can the authors provide a comparison? (not included in rating)

[1] Moon, S., Madotto, A., Lin, Z., Nagarajan, T., Smith, M., Jain, S., Yeh, C., Murugesan, P., Heidari, P., Liu, Y., Srinet, K., Damavandi, B., & Kumar, A. (2023). AnyMAL: An Efficient and Scalable Any-Modality Augmented Language Model. ArXiv, abs/2309.16058.

---

> ### Author Response · Authors · 2023-11-20
> **Response to Reviewer mDkx**
>
> Thank you so much for highly acknowledging the strengths of our work, such as 'lightweight alignment learning techniques', 'high-quality modality switching instruction-tuning dataset', and 'good and clear formulation', which means a lot to us. Following, we provide our feedback to your concerns and questions.
>
>
> * * *
> **Q1: The model relies on different pretrained models to understand different types of information, like text, images, and audio. The quality of pretraining may directly impact how well the model performs its tasks.**
> **A:** Here, we want to clarify that we actually did not utilize many different pre-trained modes to encode each individual modality. In our practice, we take advantage of a unified solution, i.e., ImageBind, to encode and project different types of information. With ImageBind, we are spared from managing many numbers of heterogeneous modal encoders. Furthermore, ImageBind is featured as a SoTA model to effectively understand six modalities. Finally, the model architect presented in this paper facilitates easy and cost-effective extension to powerful pre-trained models, thereby enhancing the understanding capabilities of different modalities.
>
>
> * * *
> **Q2: What is the size of parameters when tuning the model for each modality?**
> **A:** Actually, we have clearly stated the information about  the size of tunable parameters for each modality in Table 1 of  the paper. Specifically, the parameter size for the input alignment module is 4M, while the parameter sizes for the output alignment modules corresponding to image, audio, and video modalities are 31M, 31M, and 32M, respectively.
> |Modality | Input Projection Param.|Output Projection Param. |Total |
> | :-----: | :----: |  :-----: | :----: |
> |Text | - | - | - |
> | Image | 4M | 31M | 35M|
> | Audio| 4M | 31M | 35M|
> | Video| 4M | 32M | 36M|
>
> * * *
> **Q3: The work in [1] may be related, can the authors provide a comparison? (not included in rating)**
> **A:** Thanks for your suggestion. Truth is, we have checked the reference work [1], and as of now, the checkpoint has not been released. Thus, unfortunately, we are unable to provide a comparison at this moment. We are, however, actively monitoring the progress of that work, and once the checkpoint is released, we will supplement our paper with the subsequent comparative results. Meanwhile, we have included additional experiments, as detailed in reviewer #ijPL's rebuttal, where our results consistently demonstrate comparability or even superiority.

---

### Official Review · Reviewer_NwS3 · 2023-10-29

**Soundness:** 3 good
**Presentation:** 4 excellent
**Contribution:** 3 good
**Rating:** 5
**Confidence:** 4

**Summary:**

This paper proposes a unified framework that enables any-to-any generation. Specifically, it receives inputs from multiple modalities, such as text, audio and video. Furthermore, it leverages off-the-shelf LLMs and diffusion models, which enables efficient training. They propose to leverage special tokens that encodes the semantics, and then feed this to the diffusion model to act as the conditional input. The paper shows promising results on the evaluated benchmarks.

**Strengths:**

This paper proposes a novel approach to enable any-to-any generation by integrating off-the-shelf diffusion models with LLMs. The proposed approach to align the semantic tokens with outputs from text encoders of diffusion models seems efficient. The results look promising.

**Weaknesses:**

The major concern I have regarding this paper is the training object during alignment, which is to align the semantic tokens with outputs from text encoders of diffusion models. This seems reasonable at first, but if the objective is to "match the semantics token with textual captions' representations from the text encoders of diffusion models", why not just directly use the diffusion model's text encoder to encode the textual captions? More specifically, why not just let the LLM output a caption according to some fixed format, and extract the caption, then feed it to the diffusion model for generation? I think it would be a more direct approach and probably will enable better performance. Unfortunately, I do not find comparison with this simple alternative in the paper.

**Questions:**

Please see weakness.

---

> ### Author Response · Authors · 2023-11-20
> **Response to Reviewer NwS3**
>
> Thank you sincerely for your valuable time and comments. We've conducted additional experiments and tried every effort to address your concerns. We hope you'll reconsider your evaluation if you find our responses are effective.
>
> * * *
>
> ***(2-1)***
>
> **Q: The major concern I have regarding this paper is the training object during alignment, which is to align the semantic tokens with outputs from text encoders of diffusion models. This seems reasonable at first, but if the objective is to "match the semantics token with textual captions' representations from the text encoders of diffusion models", why not just directly use the diffusion model's text encoder to encode the textual captions? More specifically, why not just let the LLM output a caption according to some fixed format, and extract the caption, then feed it to the diffusion model for generation? I think it would be a more direct approach and probably will enable better performance. Unfortunately, I do not find comparison with this simple alternative in the paper.**
>
> **A:** Thanks for your comment.
> Actually, the approach you mentioned here that **LLM outputs textual captions and feeds to the follow-up diffusion models for generation**, is one type of prior existing methods for reaching the goal of (nearly) any-to-any MM-LLM systems, where LLMs are straightfowardly combined with external tools by connecting the LLMs with other modules with explicit textual instructions (e.g., captions or descriptions of content to generate) as the messenger. They essentially have a pipeline architecture. Representative models include Visual-ChatGPT and HuggingGPT. However, as we emphasized in the paper, there can be at least two limitations in such an approach.
>
> **Firstly**, the information transfer between different modules is entirely based on discrete texts produced by the LLM, where the cascade process inevitably introduces noise and propagates errors. This is a problem rooted in the core of different modalities, i.e., there is an inherent gap that persists between language and other modalities that cannot be eliminated. Utilizing textual captions as intermediate representations runs the risk of overlooking modality-specific features when expressing non-linguistic modalities solely through language. In other words, relying on textual captions as an intermediate representation may create a bottleneck, especially when handling complex or implicit input instructions. For example, prompting the pipeline system to generate a picture of a dog might result in a simple caption like ‘a dog running in a park’  from the LLM. However, we expect the model to generate images with much more detailed and intricate scenes more than just the ‘dog’ and a ‘park’ only, but also include vivid background details like ‘a playground with green grass, with many people walking by’, and the dog itself ‘holding a frisbee in the mouth’. All these can highlight the potential limitations of simply using textual captions as an intermediate step.
>
> **Secondly**, the entire system only leverages existing pre-trained tools for inference only, limiting its capabilities of content understanding and multimodal generation, especially in interpreting intricate and implicit user instructions. With the continual representations of special signal tokens as the messenger between different modules, we end-to-end unified MM-LLM system in the true sense. The full system can be updated throughout the whole system, which thus helps to have a much stronger capability in better understanding complex user instructions and generating content more accurately and in high quality.

---

> ### Author Response · Authors · 2023-11-20
> **Response to Reviewer NwS3**
>
> ***(2-2)***
>
> To support the above claim, during the rebuttal period, we conducted experiments to compare with such a pipeline-style baseline. We consider two types of comparisons: 1) Visual-ChatGPT and HuggingGPT, which are existing systems that have free open access; 2) NExT-GPT variant with captions as the messenger (which we mark as NExT-GPT-caption). To implement NExT-GPT-caption, the captions directly generated by LLM will be fed into the following generation models, instead of using the soft representations of the signal tokens. As Visual-ChatGPT only supports image generation, we here consider the evaluation on the Text-to-Text&Image setting.
>
> To evaluate if the system really or how well understands the input and generates output content (response text + image), we perform the human evaluation. For constructing the testing data, we first leverage GPT-4 to synthesize 1) 100 **simple** instructions (e.g., involving short and simple semantic content) that can explicitly prompt MM-LLMs to generate images, and 2) 100 **complex** instructions (e.g., involving intricate and semantically-rich scenes) that require implicit reasoning ability to generate image content. Then, the synthesized instructions are fed into the models to generate the response text + image content. Subsequently, five unbiased volunteers evaluate the generated results under three aspects, 1) **Instruction following**, identifying, among the four models, which of the generated text+image accurately responded to the input instructions, 2) **Rationality**, determining which of the generated images adhered to the input instructions, 3) **Quality**, evaluating which of the generated images exhibited the highest quality.
>
> The evaluation results are shown in Table 6&7, where we can notice the interesting observation. On the simple instructions, mostly these four models perform at similar levels. This means the impacts could be quite limited whether we take a pipeline modeling or end-to-end system on the comparatively simple user inputs. But on complex instructions, ours performs significantly better than two existing systems and NExT-GPT-caption in terms of the instruction-following capability and image generation quality. Notably, a notable degradation in the quality of generated images is observed when captions are utilized as messengers compared to the instruction-following performances. This highlights the inadequacy of captions in conveying the necessary information for generating complex images.
>
>
>
> **Table 6**: Human Evaluation (1-100 scale, results are on average) of NExT-GPT on **simple instructions** in comparison with pipeline baselines that directly generate captions for downstream generation models. The results are.
> | Model |Instruction Following |Rationality|Quality|
> | :-----: | :----: |  :-----: | :----: |
> |HuggingGPT | 94  | 92  | 87  |
> |Visual-ChatGPT | 94  | 90 | 86 |
> |NExT-GPT-caption | 93 | 90 | 81 |
> |NExT-GPT | 95 | 92 | 89 |
>
>
> **Table 7**: The results on **complex instructions**.
> | Model |Instruction Following |Rationality|Quality|
> | :-----: | :----: |  :-----: | :----: |
> |HuggingGPT |  82 |74 | 73|
> |Visual-ChatGPT | 84 | 76 | 72|
> |NExT-GPT-caption |  76 | 68 | 65 |
> |NExT-GPT |  84 | 83 | 80|

---

> > ### Comment · Reviewer_NwS3 · 2023-11-21
> > **Response to Authors**
> >
> > Thank you very much for the additional experiments. I am aware that the authors updated their experiment table in the paper, however, no visualization of the results are provided to showcase the difference between pipeline method and the proposed Next-GPT. It'd be great if the author could update the paper with a few illustrations, especially for the experiments with complex instructions, I am very curious what kind of images would cause the stable diffusion to encounter difficulty for generation with pure caption, but succeed with the help of LLM token embedding.
> >
> > Looking forward to further updates from the authors.

---

> > > ### Author Response · Authors · 2023-11-22
> > > **Re-Response to Reviewer NwS3**
> > >
> > > Thanks for your active response and the interests. To answer this, we have just updated the PDF.
> > > In `Appendix H. Case Study on Pipeline-style vs. End-to-end Unification` we added several examples of comparisons between these systems as in Figure 6, 7 and 8.
> > >
> > >
> > > Figure 6 presents the case of image generation from a simple input user instruction; while Figure 7 and 8 present two cases of image generation from comparatively complex input user instructions.
> > > On the simple one, all generated image content from both pipeline-style and end-to-end (ours) systems seem correct and coincide with the input prompt.
> > > However, when handling the complex instructions, as seen in Figure 7 and 8, the generated image content can be wrong and biased to the user intention.
> > > The problems are rooted in the core of different modalities, i.e., there are inherent gaps between language and visual modalities that cannot be eliminated.
> > > Here are two representative attributes: **the numeration of vision** (cf. Figure 6) and **the visual-spatial relational semantics** (cf. Figure 7), which could be hard to (or even cannot) be expressed by the intermediate captions conveniently.
> > > Utilizing textual captions as intermediate representations runs the risk of overlooking these modality-specific features when expressing non-linguistic (e.g., visual) modalities solely through language.
> > >
> > >
> > >
> > > By the way, we kindly note a fact that, with the intermediate captions produced from the pipeline-style systems in Figure 7 and 8, the Stable Diffusion model just has difficulty in accurately understanding the vision numeration and visual-spatial relation and generating correct answers, i.e., they are the problems
> > > inherent to the Stable Diffusion model itself, and Stable Diffusion alone is tricky to overcome.
> > > Most recent work tries to solve this issue by integrating the vision-specific features into the Stable Diffusion [1,2] via additional feature engineering.
> > > But, in our NExT-GPT with an end-to-end solution, the implicit modality signal token embeddings that carry rich modality-specific features of non-linguistic will be naturally encoded and passed to the downstream modules (e.g., Stable Diffusion), without any further external effort.
> > >
> > > [1] LayoutLLM-T2I: Eliciting Layout Guidance from LLM for Text-to-Image Generation
> > > [2] LayoutGPT: Compositional Visual Planning and Generation with Large Language Models

---

> ### Author Response · Authors · 2023-11-23
> **Eager to your reponse**
>
> Dear Reviewer,
>
> We would like to thank you again for your efforts and feedback. We have further explained about your concerns. We kindly hope that you can take some time to check our response and re-evaluate our paper based on our replies. If you have any further concerns or questions, please do not hesitate to let us know. We will be happy to address them promptly.
>
> Best

---

### Official Review · Reviewer_ijPL · 2023-10-30

**Soundness:** 3 good
**Presentation:** 3 good
**Contribution:** 2 fair
**Rating:** 5
**Confidence:** 5

**Summary:**

This paper introduces NExT-GPT, an end-to-end, multi-purpose, multi-modal Language Learning Model (MM-LLM) capable of generating text, images, audio, and video. The system is designed to be efficient, utilizing a small quantity of parameters. Furthermore, a multimodal instruction dataset named MosIT is presented, which facilitates cross-modal understanding and content generation.

**Strengths:**

1. The system architecture is compact and includes multiple decoders for text, video, audio, and image generation, making it straightforward to implement.
2. The generation process is end-to-end and does not require initial text generation.

**Weaknesses:**

1. The quality of the generation output is primarily dependent on the pre-trained generation modules. If these modules are flawed or produce errors, the system cannot rectify these issues. For instance, if in image generation, stable diffusion struggles with accurately rendering certain elements (e.g., the number of human fingers), NExT-GPT would not be able to produce an accurate output, irrespective of its understanding of the instruction.
2. The evaluation strategy appears questionable. It seems that the NExT-GPT model used in the evaluation was fine-tuned on individual datasets, which may not accurately reflect the effectiveness of the proposed method.
3. What would be the results if the model was trained on a mixture of the proposed MosIT dataset and benchmark datasets, and then evaluated on the benchmarks? Additionally, in a multimodal language model, text generation is crucial. It would be interesting to know how the system performs on recent benchmarks such as MME [1], MMBench [2], and SEEDBench [3].
4. The qualitative comparison provided lacks thoroughness. The paper only presents a few demonstrations and fails to provide comparisons with other MLLMs, including InstructBLIP [4], LLaVA [5], mPLUG-Owl [6] for text generation, and DreamLLM [7] and EMU [8] for conditioned image generation.

**Questions:**

see weaknesses

---

> ### Author Response · Authors · 2023-11-20
> **Response to Reviewer ijPL**
>
> We sincerely appreciate your valuable constructive feedback, which will surely enhance our work. We have conducted additional experiments and made every effort to address your concerns. Whenever you find we have successfully addressed your worries, please  reconsider your evaluation, thanks.
>
> ***(3-1)***
>
> * * *
> **Q1: The quality of the generation output is primarily dependent on the pre-trained generation modules. If these modules are flawed or produce errors, the system cannot rectify these issues. For instance, if in image generation, stable diffusion struggles with accurately rendering certain elements (e.g., the number of human fingers), NExT-GPT would not be able to produce an accurate output, irrespective of its understanding of the instruction.**
>
> **A:** We would like to clarify three aspects.
>
> **Firstly**, the core target or original intention of this work is to build a more unified general-purpose AI agent. That is, we do not intend to deliberately address the specific issues that individual SoTA modules (e.g., diffusion models) may face, which may quite be beyond our scope. That being said, we would like to emphasize that, to ensure obtaining an optimal system, we carefully selected all the backbone modules that are the current SoTA systems of the kind in the community, such as stable diffusion for image generation, zerocope for video generation, or AudioLDM for audio generation.
>
> **Secondly**, to our best knowledge, the most commonly adopted approach for building big LLM systems is to integrate well-established strong-performing systems. Leveraging existing pre-trained generation modules is both an effective (lowest cost) and efficient (rapid construction) method in developing unified MM-LLMs capable of delivering content in any modality, avoiding training a system from scratch, especially when the system is very huge with a big cost.
>
> **Thirdly**, the model architecture presented in this work facilitates easy and cost-effective extension to powerful incoming generation modules, potentially enabling more high-quality synthesis of image, video, or audio and beyond.
>
>
> * * *
> **Q2: The evaluation strategy appears questionable. It seems that the NExT-GPT model used in the evaluation was fine-tuned on individual datasets, which may not accurately reflect the effectiveness of the proposed method.**
>
> **A:**  Our system introduces a novel any-to-any task format, for which currently no dedicated benchmark is designed for evaluation. Under such a circumstance, when expecting to quantify the effectiveness of our model, there is no better way only can we conduct evaluations on existing benchmark datasets of the traditional individual cross-model or multimodal tasks (e.g., text-to-image, video-to-text), comparing with baselines. It is worth noticing that **we strictly adhere to the experimental settings used in the baselines of each data to ensure a fair comparison, including data splitting and fine-tuning/zero-shot setups**. While lacking a specific any-to-any multimodal benchmark (evaluation set), we are planning to further leverage our proposed MosIT dataset for this, e.g., splitting it into training and testing sets, to assess the performance of any-to-any models.
>
> Our model achieved the best performance on the majority of tasks compared to baselines, including *Audio-to-text generation*, *Image-to-text generation*, *Video-to-text generation*, and *Text-conditioned audio editing*, and also secured performances on par with SoTA baselines in other rest tasks such as *Text-to-image generation*, *Text-to-audio generation*, and *Text-conditioned image/video editing*. Notably, even in a zero-shot scenario, particularly in the Text-to-video generation task, our model surpasses the SoTA baseline.
> It is very important to emphasize a factual point that, as our goal is to build a general-purpose MM-LLM, we run on the downstream in-domain datasets to exhibit that our system indeed can perform well on existing tasks. Thus we didn’t do much task-specific optimizations and fine-tuning of our system the way the baselines do on these in-domain datasets, which actually leaves very ample room for further huge improvement to those benchmarks. In what follows, we will carefully fine-tune our system on those datasets, and it is quite promising to see our system outperforming all the existing SoTA baselines. By the way, we also plan to expand our evaluation to more benchmarks encompassing a broader spectrum of tasks, further validating our system’s advance.

---

> ### Author Response · Authors · 2023-11-20
> **Response to Reviewer ijPL**
>
> ***(3-2)***
>
> * * *
> **Q3: What would be the results if the model was trained on a mixture of the proposed MosIT dataset and benchmark datasets, and then evaluated on the benchmarks?**
>
> **A:**  To answer your question, we first would like to show how our system is trained. NExT-GPT roughly considers four stages of the training/learning process:
> - **Stage-1: Encoding-size Alignment Learning**. The input projection layer is trained by adopting an `X-to-Text` generation task on `Webvid-2M`, `CC3M`, and `AudioCaps` dataset.
> - **Stage-2: Decoding-side Alignment Learning**. The three output projection layers are optimized on  `Webvid-2M`, `CC3M`, and `AudioCaps`.
> - **Stage-3: End-to-end Instruction-Tuning**. We optimized the whole NExT-GPT using instruction-following datasets, including `Text+X → Text` Dataset, `Text → Text+X` Dataset, and `MosIT` dataset. This step is to enhance the ability of LLM to understand the user input’s instructions.
> - **Stage-4: In-domain Fine-tuning on Task-specific Training Dataset**. NExT-GPT is fine-tuned on the specific in-domain dataset of a specific task, after which the system will be evaluated on the testing set.
>
> It is worth noting that Stage-3, involving instructive tuning with MosIT data, is designed to enhance the model’s proficiency in performing complex, multi-turn, modality-switching interactions. However, in Stage-4, when evaluating on traditional benchmarks, modality alignment without intricate instruction understanding is required essentially. Without instructive tuning with MosIT, the expected impact on performance during traditional benchmark evaluation could be limited or minimal.
>
> With above discussion, for your question about *`the results if the model was trained on a mixture of the proposed MosIT dataset and benchmark datasets`*, as the objectives of the benchmark datasets and MosIT dataset are quite distinct, mixing them probably introduces the risk of interference of feature patterns, where the information learned from in-domain task-specific dataset might be disrupted by instructive tuning data. Hence, theoretically and practically, it is likely that such a mixture of training data may lead to a degradation in performance.

---

> ### Author Response · Authors · 2023-11-20
> **Response to Reviewer ijPL**
>
> ***(3-3)***
>
> ***
> **Q4: Additionally, in a multimodal language model, text generation is crucial. It would be interesting to know how the system performs on recent benchmarks such as MME [1], MMBench [2], and SEEDBench [3].**
> **A:**  During the rebuttal period, we have done the experiments as required here. Following we can present the evaluation results for MME, MMBench, and SEEDBench in Table 1, 2 and 3.
>
> **Table 1**: Evaluation results (%) on MME.
> | Model | Existence | Count | Color | Poster | Celebrity | Scene | Commonsense Reasoning  | Numerical Calculation  | Text Translation|
> | :-----: | :----: | :----: | :----: | :-----: | :----: | :----: | :----: |:----: | :----: |
> | LLaVA(7B) | 50  | 50.00 | 55.00 | 50.00 | 48.82 | 50.00 | 57.14 | 50.00 | 57.50 |
> | InstructBLIP(flant5xxl) | **185** | **143.33** | 153.33 | 123.81 | 101.18 | 153.00 | **129.29** | 40.00 | 65.00 |
> |mPLUG-Owl(7B)  | 120 | 50.00 | 55.00 | **136.5** | 100.29 | 135.50 | 78.57  | 60.00 | **80.00** |
> |NExT-GPT(7B)  | 180 | 96.67 | **156.67** | 110.00 | **103.00** | **156.25** | 116.14 | **62.50** | 65.50 |
>
>
> **Table 2**: CircularEval results on MMBench test set (L-2 abilities), including Logical Reasoning (LR), Attribute Reasoning (AR), Relation Reasoning (RR), Fine-grained Perception (Cross Instance) (FP-C), Fine-grained Perception (Single Instance) (FP-S), and Coarse Perception (CP).
> | Model | Overall | LR  | AR  | RR  | FP-S  | FP-C  | CP  |
> | :-----: | :----: | :----: | :----: | :-----: | :----: | :----: | :----: |
> | LLaVA(7B)| 36.2 | 15.9 | 53.6 | 28.6 | 41.8 | 20.0 | 40.4 |
> | InstructBLIP(7B) | 33.9 | 21.6 | 47.4 | 22.5 | 33.0 | 24.4 | 41.1 |
> | mPLUG-Owl(7B)  | 46.6 | 19.9 | 56.1 | **39.0** | **53.0** | 26.8 | 59.4 |
> | NExT-GPT(7B)  | **48.0**  | **22.1** | **60.5** | 33.6 | 46.8 | **30.7** | **60.6** |
>
>
> **Table 3**: SEEDBench evaluation results with accuracy (%) as the primary metric for each task.
> | Model | Overall | Image  | Video  |
> | :-----: | :----: | :----: | :----: |
> | InstructBLIP(7B)| 53.4 | 58.8 | 38.1|
> | mPLUG-Owl(7B)  | 34 | 37.9 | 23.0|
> |NExT-GPT(7B) | **54.4** | **59.2** | **39.4**|
>
>
> Observing the results, our model mostly achieves better text generation performance than the comparing baseline MM-LLMs the reviewer suggested here. But, it is worth highlighting that these benchmarks are quite limited to text generation, whereas our model, pioneering any-to-any generation, extends its capabilities beyond text. In addition to excelling in text responses, our model significantly contributes by proficiently generating diverse content, including image, video, and audio. This broader spectrum of generation capabilities sets our model apart from the benchmarks mentioned. As our next step, we will extend the MM-LLM leaderboard of any-to-any generation based on our MosIT data.
>
>
> * * *
> **Q5: The qualitative comparison provided lacks thoroughness. The paper only presents a few demonstrations and fails to provide comparisons with other MLLMs, including InstructBLIP [4], LLaVA [5], mPLUG-Owl [6] for text generation, and DreamLLM [7] and EMU [8] for conditioned image generation.**
> **A:**  For this question, we also provide the experimental results, by comparing NExT-GPT with other MLLMs on NoCaps, Flicker 30K, and COCO datasets (under the fair zero-shot setting), and also conditioned image generation results compared with DreamLLM and EMU. We report the CIDEr score for text generation and FID for image generation. From the results, we see that the conclusions drawn in the paper still hold water.
>
>
> **Table 4**: Zero-shot evaluation of image-to-text generation with CIDEr score (the higher the better). Results marked with * are copied from the original paper.
> | Model | NoCaps | Flickr 30K  | COCO  |
> | :-----: | :----: | :----: | :----: |
> |InstructBLIP(7B) | 123.1* | 82.4* | 102.2*|
> |LLaVA(7B) | 120.7 | 82.7 | -|
> |mPLUG-Owl(7B) | 117.0 | 80.3 | 119.3|
> |EMU(13B) | - | - | 117.7*|
> |DreamLLM(7B)| - | - | 115.4*|
> |NExT-GPT(7B) | **123.6** | **84.5** | **124.9**|
>
>
> **Table 5**: Zero-shot evaluation of Text-to-Image Generation (COCO) with FID (the lower the better) score. Models pre-trained with LION (2B) data are denoted with *.
> | Model | FID |
> | :-----: | :----: |
> |EMU*(13B) | 11.66|
> |DreamLLM*(7B)| **8.46** |
> |NExT-GPT(7B) | 13.85|
> |NExT-GPT*(7B) | 8.62|

---

> ### Author Response · Authors · 2023-11-23
> **Eager to your reponse**
>
> Dear Reviewer,
>
> We would like to thank you again for your efforts and feedback. We have explained in detail about the concerns you raised before.
> We kindly hope that you can take some time to check our response and re-evaluate our paper based on our replies. If you have any further concerns or questions, please do not hesitate to let us know. We will be happy to address them promptly.
>
> Best

---

### Official Review · Reviewer_7uQJ · 2023-11-07

**Soundness:** 3 good
**Presentation:** 4 excellent
**Contribution:** 3 good
**Rating:** 6
**Confidence:** 4

**Summary:**

This paper proposes a multi-modal LLM, called any-to-any MM-LLM, to extend the multi-modality of LLM to a state where there is no limitation on the input and output modality combinations. To achieve this goal, the authors (1) propose a lightweight alignment learning technique to achieve en effective semantic alignment across different modalities with limited trainable parameters and (2) annotate a modality-switching instruction tuning dataset. The displayed results and visualizations suggest the promising performance of the tuned any-to-any MM-LLM.

**Strengths:**

- Extending the multi-modal LLMs free of limitation on the input/output modalities is an important research question that can facilitate a wider range of applications.
- The introduced dataset, if made publically available, would be a good contribution to the community.
- Various evaluation benchmarks are used to benchmark the proposed model with existing solutions.
- The writing is clean and easy to follow

**Weaknesses:**

- The proposed alignment learning technique is a bit naive and does not consider much about the challenge introduced by the any-to-any modality, such as how to balance the performance across different modalities.
- Although introducing contents from different modalities during tuning is considered to improve the overall performance of the model, in the experiment section, it seems introducing these additional modalities actually leads to worse performance on benchmarking datasets. Does this indicate the alignment technique is not effective enough as expected?

**Questions:**

Will the pretrained model and dataset be released to the public?

**Details Of Ethics Concerns:**

The proposed datasets's content may need a deeper look from experts to check its content. And the content generated by the model may need further checking to make sure there are no harmful contents generated.

---

> ### Author Response · Authors · 2023-11-20
> **Response to Reviewer 7uQJ**
>
> Thank you for your valuable feedback, especially for the recognition of our strengths, such as 'important research',  'valuable datasets', and 'well-structured and easy to follow'. Below, we show you the point-to-point responses, which we hope can address your concerns.
>
> * * *
> **Q1: The proposed alignment learning technique ... different modalities.**
>
> **A:**  Thanks for your insights; it’s a good point. But here, we would like to clarify a few points.
>
> **Firstly**, our proposed alignment learning techniques are featured as lightweight and cost-effective, facilitating rapid iteration of computational-centric model training.
>
> **Secondly**, we have indeed considered the challenges introduced by the ”any-to-any modality” paradigm more or less, including the performance balance issue. For example, utilizing a unified encoder (i.e., ImageBind) to perceive the non-language modalities, instead of separate encodings, helps mitigate imbalances between models. Additionally, we have designed three independent output projection layers to align LLM with downstream generation systems, alleviating the interference between modalities and maintaining a balance.
>
> **Thirdly**, in contrast to balancing the performance across different modalities, we emphasize that it can be much more prioritized to maximize the performance of individual existing encoder and decoder used in our system, achieving fully optimal performance of each modality.
>
> Admittedly, there is ample room for improvement of more advanced alignment learning, and we appreciate your suggestions and we will explore further in the future.
>
> * * *
> **Q2: Although introducing ... as expected?**
>
> **A:** Thanks. Here, first of all, we would like to clarify a misunderstanding that, our paper never (or intended to) claimed that **introducing other modalities enhances the performance of the specific modality**. Our focus always lies in constructing a unified agent, i.e., NExT-GPT, to mimic the complementary nature of different modalities in real-world scenarios. The proposed alignment tuning is designed to project different modalities into a shared semantic space, achieving such a unified agent where different modalities are complementary, rather than enhancing individual modality performance. We would like to note that, the performance mutual enhancement of multimodal systems is mostly observed and expected in tasks involving multimodal inputs that different modalities refer to shared semantic meaning, which however is not the primary focus of our system.
>
> In addition, regarding the concern about *`leads to worse performance on benchmarking datasets`*, experimental results yet indicate that our system outperforms the baselines in tasks ranging from text-to-video to audio-to-text, image-to-text, and video-to-text generation. For the remaining evaluation tasks, our model still secured performances on par with the best-performing baselines, as shown in the tables. We kindly note that, most SoTA baseline systems are deliberately created for addressing that particular benchmarks and tasks which are well fine-tuned on the datasets for achieving a very high score, while our systems focus more on the overall (general-purpose) any-to-any capabilities. We want to claim a fact that, we didn’t do much task-specific optimizations and fine-tuning of our system as the baselines do, which actually leaves ample room for further huge improvement of benchmarks.
>
>
> * * *
> **Q3: Will the pretrained model and dataset be released to the public?**
>
> **A:**  Yes, all our pre-trained model (also will be consistently updated with new versions) and datasets will be released to the public to facilitate the follow-up research.
>
> * * *
> **Q4: Ethics concerns about the content generated by the model or the dataset content**
>
> **A:** Thanks for bringing this attention to us. In addressing ethical considerations related to the content generated by Next-GPT, several key measures are implemented.
>
> **In the model aspect**, Next-GPT is built upon existing well-established models, where we take full advantage of their inherent safety mechanisms to ensure the quality of generation content. Then, we emphasize user awareness by advising adherence to usage guidelines (e.g., interpreting the generated content with caution) and strictly limiting applications to academic research rather than commercial purposes.
>
> **In the data aspect**, concerning data creation, we take meticulous measures to protect privacy, e.g., by removing or obfuscating personally identifiable information. We remain vigilant in mitigating bias during dataset collection, striving for representativeness to avoid favoring or disadvantaging any specific group or perspective. It is important to note that we possess the necessary licenses to collect data. And for data distribution to researchers  and individuals, adherence to the stipulated license is mandatory.

---

> > ### Author Response · Authors · 2023-11-23
> > **Eager to your reponse**
> >
> > Dear Reviewer,
> >
> > We would like to thank you again for your efforts and feedback. We have explained in detail about the concerns you raised before.
> > We kindly hope that you can take some time to check our response and re-evaluate our paper based on our replies. If you have any further concerns or questions, please do not hesitate to let us know. We will be happy to address them promptly.
> >
> > Best

---

### Author Response · Authors · 2023-11-20
**General Response to All Reviewers**

## Dear reviewers,

Thanks for all of your time in providing detailed and constructive comments. Your feedback is important to assist us in enhancing the quality of our work, and we are fully committed to incorporating your suggestions into our revision process. At this juncture, we would like to re-emphasize the significance of this work:
1. We introduce an end-to-end unified MM-LLM, capable of semantic understanding and reasoning across various modalities;
2. We design lightweight alignment learning techniques that efficiently require only 1% of parameter optimization;
3. We present a meticulously annotated high-quality modality-switching instruction tuning dataset.

----

 Additionally, we would like to address some common concerns raised by the reviewers:

**Concern-1: NExT-GPT relies much on existing pre-trained models, where the performance of  NExT-GPT depends on the latter, rather than enhancing the generation of multimodal contents.**
- **A:** **Firstly**, our primary goal in this paper is to construct a unified agent effectively, prioritizing overall capability, instead of deliberately addressing the issues in the existing generative model (e.g., diffusions) used in NExT-GPT.
**Secondly**, leveraging existing well-established models has long been a commonly-adopted and effective approach for rapidly building unified systems.
**Thirdly**, our framework advances in good modularization, where the extensibility allows for easily integrating potentially more powerful and advanced incoming modules and models (e.g., replacing Stable-Diffusion with SDXL).

**Concern-2: Not beating all comparing baselines on all benchmarks, which questions the efficacy of the proposed alignment learning strategies**.
- **A:** **Firstly**, we want to clarify that our model outperforms the best-performing baselines on the majority of tasks, and the performance on the remaining tasks is on par with the SoTA baselines.
**Secondly**, we kindly note that most SoTA baseline systems we included in the paper are deliberately designed to address particular benchmarks and tasks. They are intentionally well fine-tuned on those in-domain datasets for achieving very high scores, while our NExT-GPT system focuses more on the general-purpose any-to-any capabilities. We would like to claim that, we did not do much task-specific optimizations and fine-tuning of our system as the baselines do, which actually leaves ample room for further huge improvement on benchmarks to be the SoTA.

----


Again, we express our gratitude for the thoughtful reviews, and we have thoroughly reviewed our paper, again polished the paper (as uploaded in the revised PDF), performed additional experiments, and then crafted a comprehensive response as follows.
We hope that our rebuttal adequately addresses your concerns. We are also eager to see reviewers #ijPL and #NwS3 can change the slightly negative ratings into positive ones when our responses can effectively address the concerns, and look forward to your response to our rebuttal.

----

For answering the questions of reviewers #ijPL and #NwS3, we added new experiment results, which are further included in our revision. Here, we specify the following modifications in the revised PDF:
1.  In the Experiment section, we extend our evaluations under more settings as required by reviewers, including 1) zero-shot evaluation  (refer to Table 11) on text-to-image generation and image-to-text generation, 2) evaluation on several multimodal LLM benchmark (refer to Table 12&13) and 3) evaluation on pipeline vs end-to-end MM-LLMs (refer to Table 14).
2. We add a section of Appendix: *H. Case Study on Pipeline-style vs. End-to-end Unification*.
2. We move the *Human Evaluation on Complex Any-to-any QA*  and *Limitation and Future Work* to the appendix.
3. We polish the whole paper and improve clarity and coherence.

---

### Meta-Review · Area_Chair_VaGd · 2023-12-11

**Metareview:**

The paper introduces an any-to-any multimodal LLM, where modality alignment is performed not only in the input as in previous approaches, but also in the output (between the LLM and diffusion decoders of different modalities). The reviewers raised concerns about the over-reliance on pre-trained models and unclear effectiveness of the proposed approach. The authors prepared a very extensive response, with many new experiments, including more baselines, performance on additional benchmarks, and qualitative results (We thank the authors for their effort!). The final scores were mixed, with one reviewer recommending acceptance, two borderline rejects, and one borderline accept. As a borderline case, the paper was then discussed by the AC and SAC. They both see merits in the paper, and consider the problem of studying any-to-any multimodal models very important. However, the following remain major concerns: 1) The issue with evaluation goes beyond beating SOTA methods, as there is no standard benchmark for evaluating any-to-any models. In this respect, it is difficult to properly evaluate the proposed approach; 2) We consider legitimate the concern raised by the reviewers that the alignment techniques are a bit naive and most of the work is done by the pre-trained models — in this sense, the technical contribution is limited; 3) The authors should tone down some claims in the paper about human-level capabilities and reasoning (and anyway they would be inherited by the pre-trained models rather than the proposed method). Finally, the paper would benefit from a second round of reviews given the new experiments provided in the rebuttal. Based on these reasons, the ACs recommend rejection, but encourage the authors to further improve the paper for another top-tier conference or journal submission.

**Justification For Why Not Higher Score:**

See the three reasons stated above in the meta-review

**Justification For Why Not Lower Score:**

N/A

---

### Decision · Program_Chairs · 2024-01-16

Reject